# TEOCHAT: A LARGE VISION-LANGUAGE ASSISTANT FOR TEMPORAL EARTH OBSERVATION DATA

**Jeremy Andrew Irvin\*, Emily Ruoyu Liu, Joyce Chuyi Chen, Ines Dormoy,**
**Jinyoung Kim, Samar Khanna, Zhuo Zheng, Stefano Ermon**
Stanford University
\*Correspondence to: `jirvin16@cs.stanford.edu`

## ABSTRACT

Large vision and language assistants have enabled new capabilities for interpreting natural images. These approaches have recently been adapted to earth observation data, but they are only able to handle single image inputs, limiting their use for many real-world tasks. In this work, we develop a new vision and language assistant called TEOChat that can engage in conversations about temporal sequences of earth observation data. To train TEOChat, we curate an instruction-following dataset composed of many single image and temporal tasks including building change and damage assessment, semantic change detection, and temporal scene classification. We show that TEOChat can perform a wide variety of spatial and temporal reasoning tasks, substantially outperforming previous vision and language assistants, and even achieving comparable or better performance than several specialist models trained to perform specific tasks. Furthermore, TEOChat achieves impressive zero-shot performance on a change detection and change question answering dataset, outperforms GPT-4o and Gemini 1.5 Pro on multiple temporal tasks, and exhibits stronger single image capabilities than a comparable single image instruction-following model on scene classification, visual question answering, and captioning. We publicly release our data, model, and code at `https://github.com/ermongroup/TEOChat`.

## 1 INTRODUCTION

Many earth observation (EO) tasks require reasoning over time. For example, change detection is a widely studied task with the goal of identifying salient changes in a region using EO images at different times (Chughtai et al., 2021). Automated methods to perform change detection can greatly aid humanitarian and sustainability efforts, including planning disaster relief (Khan et al., 2023), tracking urban change (Reba & Seto, 2020), and monitoring deforestation (Mitchell et al., 2017), crops (Karmakar et al., 2024), and ecological conditions (Besson et al., 2022). Prior automated methods to detect change in EO imagery have been specialist models, constraining their use to single tasks they were explicitly trained to perform (Bai et al., 2023; Cheng et al., 2024).

Advancements in the modeling of multimodal data have enabled generalist vision-language models (VLMs) that can perform a variety of natural image interpretation tasks specified through natural language (Liu et al., 2023a). These models combine the capabilities of large language models and vision encoders, aligning visual and textual representations to enable natural, conversational interfaces for diverse tasks. This flexible interface presents a major opportunity for developing multimodal conversational agents for a variety of real-world applications, such as emergency response, where rapid and adaptable information processing is crucial.

New VLMs have gained capabilities to model temporal sequences of natural images (video) (Tang et al., 2023) and single EO images (Li et al., 2024c). However, no prior VLMs can model *temporal EO data* (left of Figure 1). We investigate the performance of Video-LLaVA (Lin et al., 2023), a strong natural image VLM that can receive images and videos as input, and GeoChat (Kuckreja et al., 2023), a strong VLM fine-tuned on *single* EO image tasks (right of Figure 1). We find Video-LLaVA generates inaccurate information, likely due to being trained on natural images and videos, whereas GeoChat only inputs single images and cannot process information across time.

| Model | Temporal | EO |
|-------|:--------:|:--:|
| LLaVA (Liu et al., 2024a) | ✗ | ✗ |
| LLaVA-1.5 (Liu et al., 2023a) | ✗ | ✗ |
| VideoChatGPT (Maaz et al., 2023) | ✓ | ✗ |
| Video-LLaVA (Lin et al., 2023) | ✓ | ✗ |
| Chat-UniVi (Jin et al., 2023) | ✓ | ✗ |
| LLaMA-VID (Li et al., 2023) | ✓ | ✗ |
| Video-LLaMA (Zhang et al., 2023b) | ✓ | ✗ |
| ST-LLM (Liu et al., 2024b) | ✓ | ✗ |
| RSGPT (Hu et al., 2023) | ✗ | ✓ |
| GeoChat (Kuckreja et al., 2023) | ✗ | ✓ |
| SkyEyeGPT (Zhan et al., 2024) | ✗ | ✓ |
| LHRS-BOT (Muhtar et al., 2024) | ✗ | ✓ |
| VHM (Pang et al., 2024) | ✗ | ✓ |
| **TEOChat (Ours)** | ✓ | ✓ |

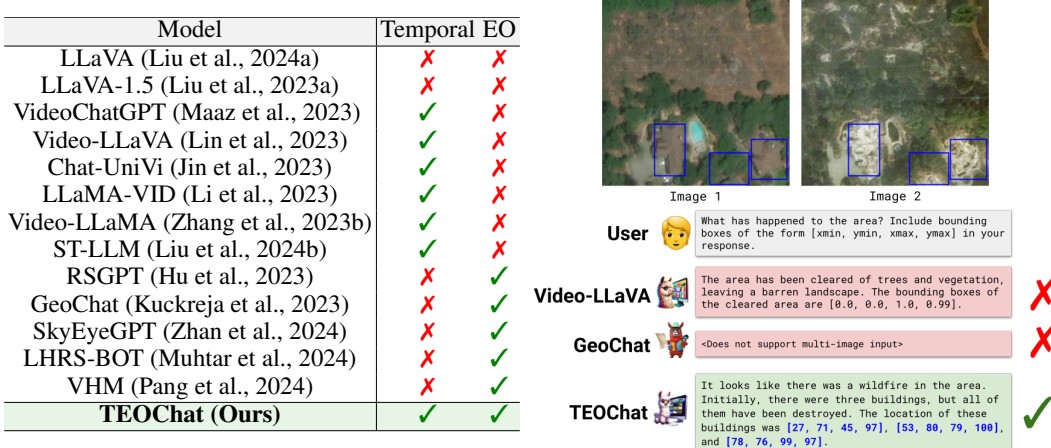

Figure 1: **Left: VLM Capabilities.** TEOChat is the first VLM for temporal EO data. **Right: Example outputs of prior VLMs.** We compare to a temporal VLM (Video-LLaVA) and a single EO image VLM (GeoChat).

We address these limitations by introducing TEOChat, the first VLM for temporal sequences of EO images. TEOChat adapts the Video-LLaVA architecture, designed originally for natural images and videos, to single and temporal sequences of EO images. To train TEOChat, we curate TEOChatlas, the first instruction-tuning dataset with instruction-following examples for temporal EO data. We construct a variety of tasks which require spatial and temporal reasoning capabilities using four EO datasets, namely fMoW (Christie et al., 2018), xBD (Gupta et al., 2019), S2Looking (Shen et al., 2021), and QFabric (Verma et al., 2021). TEOChatlas is large and diverse in task composition, sensors, and geography, making it a strong dataset for training generalizable EO VLMs.

In summary, our contributions are:

1. We develop the first vision-language model for temporal EO data called **TEOChat**. When prompted with temporal images and natural language instructions, TEOChat can perform EO tasks that require temporal reasoning, e.g. change detection. TEOChat outperforms a previous VLM for single EO images (GeoChat) and a VLM for temporal sequences of natural images (Video-LLaVA), and also rivals specialist models on multiple tasks.

2. We introduce the first temporal EO dataset for multimodal instruction-tuning called **TEOChatlas** to train TEOChat. TEOChatlas contains 245,210 temporal EO examples spanning dozens of temporal instruction-following tasks as well as 308,861 single EO examples from GeoChat_Instruct, totalling 554,071 examples and 1,244,393 images.

3. We conduct a thorough set of experimental ablations to motivate TEOChat's design. We find the key design decisions include Video-LLaVA initialization, vision-language connector fine-tuning, and the inclusion of image references between visual tokens in the prompt.

4. We demonstrate that (i) TEOChat achieves impressive zero-shot performance on an EO change detection and a change QA dataset, (ii) TEOChat outperforms two strong proprietary foundation models for modeling sequences of images (GPT-4o and Gemini-1.5 Pro), and (iii) TEOChat possesses strong single image capabilities, outperforming GeoChat on multiple zero-shot scene classification and visual question answering (VQA) tasks.

## 2 DEVELOPING A VLM FOR TEMPORAL EO DATA

We aim to develop a VLM that can input a natural language instruction and a temporal sequence of EO images, and output a natural language response. For example, given satellite images capturing an urban area before and after a hurricane, the model should respond to questions about damaged buildings or flooded regions. The vast majority of natural image VLMs for this task use a LLaVA-like architecture (Liu et al., 2024a) consisting of (i) a large language model (LLM; commonly LLaMA 2 (Touvron et al., 2023) initialized with Vicuna weights (Chiang et al., 2023)) (ii) a vision encoder (commonly a ViT (Dosovitskiy et al., 2020) pre-trained with CLIP (Radford et al., 2021)) to generate visual representations, and (iii) a vision-language connector (commonly an MLP (Liu et al., 2023a)) to align the visual representations with the language embeddings input to the LLM.

Training the VLM requires aligning language representations with visual concepts in EO images, but these concepts are not well-represented in natural image data. EO imagery is taken from a birds-eye view and captures land use and land cover changes like flooding or road construction. To align EO data with language and allow the model to respond to instructions about it, we require a dataset of instruction, EO image, and ground truth response triplets (an instruction-following dataset). The responses are used to supervise the VLM using a next-token cross-entropy loss on the response tokens (visual instruction tuning (Liu et al., 2024a)). We note that existing instruction-following EO datasets are insufficient for training a temporal EO VLM, as the tasks do not require temporal capabilities and therefore do not promote their development. We confirm this empirically by showing that GeoChat substantially underperforms our approach across all temporal reasoning tasks we test.

## 3 TEOChatlas: A Temporal EO Instruction-Following Dataset

The main obstacle to developing a temporal EO VLM is the absence of a suitable instruction-following dataset. To overcome this, we curate an instruction-following dataset of temporal EO imagery called TEOChatlas. A few key desired capabilities of the VLM induce important dataset design decisions. First, the model should follow instructions about a wide variety of tasks requiring spatial and temporal reasoning capabilities, so we construct a diverse set of instruction-following tasks (Section 3.1). Second, the model must reason over variable-length temporal sequences of EO data from different sensors, so we ensure many sensors and sequence lengths are represented (Section 3.2). Third, the model should be able to reference specific images in its input and output, and support user-friendly conversation, so we design the prompts to support these features (Section 3.3).

### 3.1 TEOChatlas: Task Composition

We curate instruction-following tasks in TEOChatlas to develop a range of model capabilities. Importantly, because our goal is to add temporal capabilities without compromising single image capabilities, we include both single image and temporal tasks in the dataset, described below.

#### 3.1.1 Single Image Tasks

To support the development of single image capabilities like object recognition and spatial reasoning, we include many single image instruction-following tasks in TEOChatlas. Specifically, we use GeoChat_Instruct (Kuckreja et al., 2023), a composition of single image instruction-following tasks including scene classification, visual question answering, referring expression, region captioning, detailed description, grounded description, and multi-turn conversation. GeoChat_Instruct was used to train the GeoChat model, which exhibits strong performance on these single image tasks. Appendix B has more detail about GeoChat_Instruct and instruction-following tasks.

#### 3.1.2 Temporal Tasks

We construct temporal instruction-following tasks using four EO datasets. We group tasks into seven categories spanning two common real-world applications, namely disaster response and urban development monitoring. Figure 2 shows example of tasks and Table 8 enumerates all of the tasks and prompts. TEOChatlas is the first instruction-following EO dataset with temporal tasks (Table 6).

**Temporal Scene Classification (TSC)** We construct an instruction-following task from a standard EO task called temporal scene classification, where the goal is to classify a sequence of satellite images into a set of predefined categories. For this task, we use the Functional Map of the World (fMoW) dataset (Christie et al., 2018), consisting of satellite image time series classified as one of 62 categories. To convert fMoW to an instruction-following task, we use a standard prompt which instructs the model to classify the image by selecting one of the 62 classes provided in the prompt.

**Change Detection (CD)** We curate instruction-following tasks for change detection, a widely studied earth vision task to identify changes in images of an area over time. We include two canonical change detection tasks, namely building damage assessment and building change detection.

For building damage assessment, we use xBD (Gupta et al., 2019), a building damage detection dataset consisting of bitemporal pre- and post-disaster images, where every building is localized and classified into four damage severity categories. Following standard practice on this dataset,

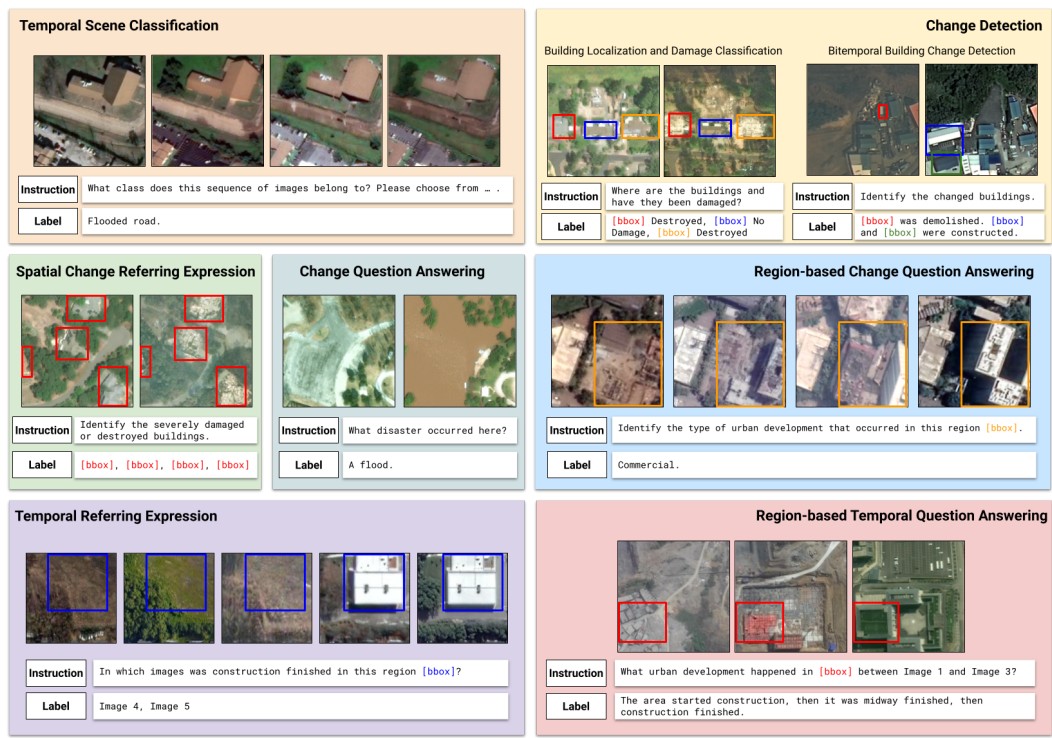

Figure 2: **Examples of instruction-following tasks in TEOChatlas.** We curate many instruction-following tasks for temporal EO data and group them into seven categories. The tasks require spatial and temporal reasoning capabilities, and span real-world applications including disaster relief and urban development monitoring.

we divide this task into two subtasks, namely building localization (*Loc.*) and building damage classification (*Dmg Cls.*). Building localization prompts the model to predict bounding boxes around all buildings in the area, representing each box as a sequence of natural language tokens of the form $[x_{\min}, y_{\min}, x_{\max}, y_{\max}]$ following Kuckreja et al. (2023). Building damage classification requires the model to classify each bounding box input in the instruction as one of the four damage categories provided in the prompt. These tasks together form a canonical semantic change detection task.

For building change detection (*Det.*), we use S2Looking (Shen et al., 2021), a building change detection dataset consisting of bitemporal, side-looking satellite images labeled with polygons indicating that a building has been constructed or demolished. We follow standard practice and task the model with outputting all changed buildings in the images. We use the same type of instruction as was used for the building localization task. This is a canonical binary change detection task.

**Spatial Change Referring Expression (SRE)** To facilitate better spatial reasoning capabilities, we create instruction-following tasks to identify the locations of specific changes which are referred to by natural language expressions. Specifically, we create tasks from xBD where the model is instructed to identify the locations of only the buildings which experienced a specific damage type. We also construct tasks from S2Looking where the model is required to detect only the constructed buildings or only the demolished buildings. These temporal tasks are analogous to the referring expressions for single images used in the GeoChat dataset (Kuckreja et al., 2023).

**Change Question Answering (QA)** To develop the model's ability to recognize change and reply to a diverse array of user questions, we construct multiple visual question answering tasks about changes between temporal images. From xBD, we create tasks such as instructing the model to output the type of disaster from the provided labels in the dataset, the location in the region (a cell of a 3×3 grid specified with top left, top center, etc.) most affected by the disaster, or a binary label indicating if there are any destroyed buildings in the area. Similarly, we add tasks from S2Looking to identify whether any buildings have been constructed or demolished in the region.

**Region-based Change Question Answering (RQA)** We create question answering tasks based on regions to improve reasoning about spatial inputs. We generate the same change question answering tasks from xBD and S2Looking, but constrain the question to a bounding box input representing a specific area within the image. We also create tasks using QFabric (Verma et al., 2021), a multi-task urban change detection dataset consisting of five image sequences each labeled with polygons representing urban change. Each polygon has a sequence-level label indicating the broad type of change (change type classification) and an image-level label indicating the specific change that occurred between adjacent timesteps (change status classification). We construct instruction-following tasks by mirroring building damage classification, instructing the model to classify the change type or change status of an input bounding box into categories included in the prompt.

**Temporal Referring Expression (TRE)** To enable the model to temporally localize change, we include tasks instructing the model to identify the specific image(s) in which a given change occurred. We use the change type labels from QFabric which indicate the type of urban development that occurred in each image. Because these labels are tied to specific regions of the images, we include a bounding box in the prompt and task the model with identifying the image(s) where the change occurred. To perform this task, the model has to model spatial inputs and generate temporal outputs.

**Region-based Temporal Question Answering (RTQA)** In order to support the development of sophisticated spatial and temporal reasoning capabilities together, the final tasks we curate for TEOChatlas instruct the model to answer questions about a given region, inputting or outputting references to images in the sequence. Specifically, we create questions about the presence of any urban development across two images within a box specified in the prompt and we generate tasks instructing the model to identify all images in the sequence where a given change type occurred.

### 3.2 TEOCHATLAS: EO IMAGE SEQUENCES

To allow the model to reason about varying input sequence lengths and sensor types, we ensure that TEOChatlas includes a diverse set of image sequence sizes and image providers. For sequence lengths, the temporal EO datasets we include consist of bitemporal (xBD and S2Looking), pentatemporal (QFabric), and multitemporal (variable length) sequences (fMoW). Due to memory constraints, we limit the sequences to a maximum of 8 images, and randomly sample 8 images without replacement from longer sequences. To increase the representation of other sequence lengths in the dataset, we also randomly sample shorter sequences from QFabric. For sensor types, the EO datasets in TEOChatlas span 8 sensors (Table 7). To increase the representation of Sentinel-2, a publicly available sensor with global coverage that is commonly used for EO tasks, we include sequences from fMoW-Sentinel (Cong et al., 2022). More detail about how we process the images is in Appendix B, and we also note that TEOChatlas has considerable geographic diversity (Figure 5).

### 3.3 TEOCHATLAS: PROMPT DESIGN

We design the prompts to enable the model to reference images and converse in a user-friendly manner. As the TRE and RTQA tasks require the model to input and output references to specific images in the sequence, we design the prompts to facilitate this. To do so, we interleave image references between the visual tokens input to the LLM (see Figure 3). This differs from prior methods for temporal sequences of natural images which simply concatenate the representations (Maaz et al., 2023; Lin et al., 2023; Zhang et al., 2023b; Jin et al., 2023; Li et al., 2023; Touvron et al., 2023).

GeoChat_Instruct depends on task tokens to help the model differentiate between tasks, for example to determine whether to include boxes in its response. We remove this requirement, and replace the tokens with short natural language specifications of the task. Examples are provided in Table 8.

Finally, we modify the system prompt to (1) specify that the input contains a sequence of satellite images and (2) randomly include the image resolution (high or low) and sensor name in the prompt to allow TEOChat to leverage that information when provided (see Appendix).

## 4 TEOCHAT

### 4.1 TEOCHAT ARCHITECTURE

We adopt a LLaVA-1.5-like architecture (Liu et al., 2023a) consisting of (i) a temporally-shared image encoder (CLIP ViT-L/14 (Radford et al., 2021)) to obtain representations of each image in

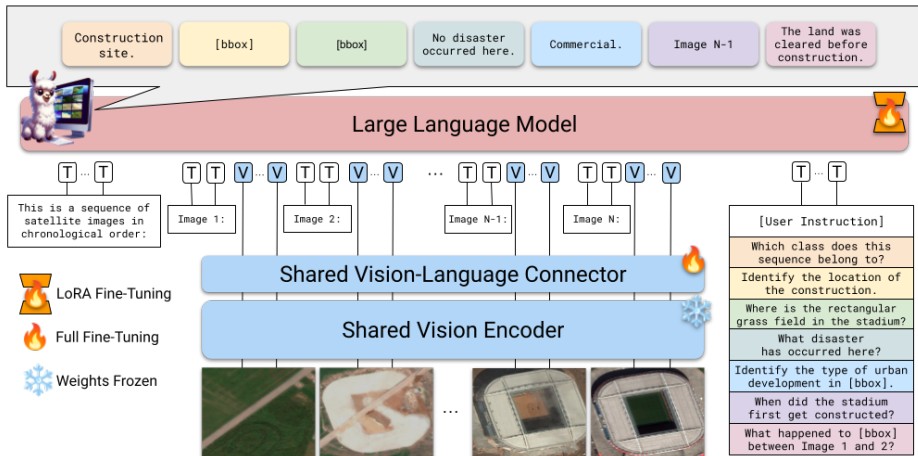

Figure 3: **Overview of TEOChat.** TEOChat inputs a temporal sequence of EO images and a user instruction, and outputs a natural language response. It can input and output regions specified through bounding boxes, and can also input and output references to specific timesteps using image identifiers. We show one example from each task category, including temporal scene classification (peach), change detection (yellow), spatial change referring expression (green), change question answering (cyan), region-based change question answering (blue), temporal referring expression (purple), and region-based temporal question answering (pink).

the sequence, (ii) a 2-layer MLP to project the visual representations to the input of the LLM, and (iii) an LLM decoder (Llama 2 (Touvron et al., 2023)) which inputs the instruction and temporal sequence of projected visual representations to generate the response (Figure 3). A recent extension of LLaVA-1.5 to handle videos called Video-LLaVA (Lin et al., 2023) uses joint natural image-video instruction-tuning by using the frozen CLIP ViT image encoder for images and a frozen Language-Bind (CLIP-aligned) video encoder for videos (Zhu et al., 2023a), while keeping the rest of the architecture the same as LLaVA-1.5. We test whether these projector and LLM weights learned on video data are better for temporal EO data by comparing Video-LLaVA initialization to LLaVA initialization, as well as freezing vs. fine-tuning the projector, in our experimental ablations.

Another key architectural decision is whether to use the image encoder or video encoder from Video-LLaVA (Lin et al., 2023) to generate visual representations for the sequence. We choose to use the image encoder for two reasons: (1) prior change detection approaches have demonstrated that Siamese encoders with weights shared across time (temporal-agnostic) are highly effective for identifying changes in sequences of EO data (Zheng et al., 2021a;c; 2022a), and (2) the video encoder was designed to input fixed sequences of 8 images. While it is possible to remove this second constraint, we observed performance degradation when using shorter sequences in preliminary experiments, likely because the video encoder was trained strictly on 8 image sequences (Zhu et al., 2023a). Replicating images in the sequence may address this but makes training and inference much slower. Altogether, our vision encoder is fast, memory efficient, and achieves strong performance.

## 4.2 TEOCHAT TRAINING DETAILS

Following prior work (Kuckreja et al., 2023), to retain the strong capabilities of the pre-trained image encoder and LLM, and to minimize memory usage during training, we freeze the vision encoder then fine-tune the LLM using Low-Rank Adaptation (LoRA) (Hu et al., 2021). To further reduce memory footprint, we use 8-bit quantization of the base LLM weights. These strategies allow us to train the large multimodal model on temporal sequences of up to 8 images using an NVIDIA A4000 GPU (16GB VRAM). We provide additional training details in Appendix Section F.

## 5 EXPERIMENTAL RESULTS

We evaluate TEOChat in a variety of settings. First, we find TEOChat demonstrates impressive temporal reasoning capabilities, outperforming two strong VLMs, one for videos (Video-LLaVA) and one for single EO images (GeoChat), and even rivals or outperforms several specialist approaches (Section 5.1). Second, through experimental ablations we show TEOChat's design is superior to

| Task | Dataset/Subtask | Specialist | Video-LLaVA | GeoChat | TEOChat-T | TEOChat |
|------|-----------------|------------|-------------|---------|-----------|---------|
| TSC | fMoW RGB (Acc.) | 65.9 | 16.6 | 59.2 | 73.5 | **75.1** |
|     | fMoW Sentinel (Acc.) | - | 4.9 | 26.3 | **45.5** | **45.5** |
| CD | xBD Loc. ($F_1$) | 85.9 | 7.0 | 7.4 | 32.9 | **38.9** |
|    | xBD Dmg Cls. ($F_1$) | 26.5 | 7.0 | 11.8 | 49.4 | **50.0** |
|    | S2Looking Det. ($F_1$) | 26.5 | 8.0 | 7.7 | 30.6 | **34.5** |
| SRE | xBD ($F_1$) | - | 3.7 | 8.7 | 18.3 | **25.1** |
|     | S2Looking ($F_1$) | - | 1.7 | 4.5 | 31.0 | **32.9** |
| QA | xBD (Acc.) | - | 30.8 | 34.0 | 89.8 | **89.9** |
|    | S2Looking (Acc.) | - | 41.5 | 51.5 | **75.4** | 73.4 |
| RQA | xBD (Acc.) | - | 57.3 | 63.6 | 93.7 | **94.0** |
|     | S2Looking (Acc.) | - | 47.8 | 67.2 | 86.7 | **90.0** |
|     | QFabric [2 images] ($F_1$) | 77.0 | 22.6 | 20.9 | 63.9 | **66.7** |
|     | QFabric [5 images] ($F_1$) | 81.6 | 26.5 | 21.9 | 72.8 | **74.3** |
| TRE | QFabric (Acc.) | - | 1.9 | - | 73.1 | **74.9** |
| RTQA | QFabric (Acc.) | - | 26.5 | - | 71.4 | **71.7** |
|      | QFabric [5 images] ($F_1$) | 54.5 | 10.5 | 12.5 | 65.2 | **66.4** |

Table 1: **Temporal task performance of TEOChat compared to other VLMs and specialists.** We bold the best generalist result on each task. TEOChat outperforms a VLM instruction-tuned on natural images and videos (Video-LLaVA) as well as a VLM instruction-tuned on single EO images (GeoChat). We note Video-LLaVA and GeoChat are not trained on these tasks. TEOChat also performs comparably to or outperforms several specialist models trained to perform specific tasks. TEOChat-T was only trained on temporal tasks.

many others (Section 5.2). Third, TEOChat achieves impressive zero-shot generalization to new temporal datasets not in TEOChatlas (Section 5.3). Fourth, we find that TEOChat outperforms two strong proprietary VLMs trained on sequences of images (Section 5.4). Fifth, we show TEOChat possesses strong single image capabilities, outperforming GeoChat on average across zero-shot scene classification, VQA, and captioning (Section 5.5). Sixth, we find TEOChat exhibits interesting instances of generalization to tasks not explicitly represented in the training set (Section 5.6).

## 5.1 COMPARISON TO OTHER VLMS AND SPECIALIST APPROACHES

**Baselines and Evaluation Metrics** We compare TEOChat to Video-LLaVA (Lin et al., 2023), GeoChat (Kuckreja et al., 2023), and specialist models. To obtain GeoChat's predictions on the temporal tasks, we perform temporal post-processing of its single image outputs (Appendix Section G). For the specialists, we use a strong self-supervised EO model adapted to fMoW (SatMAE (Cong et al., 2022)) and strong models trained on each individual dataset for the change detection tasks (a modified UNet (Ronneberger et al., 2015) from Gupta et al. (2019), FC-Siam-Diff (Daudt et al., 2018) from Shen et al. (2021), and a modified UNet from Verma et al. (2021)). We evaluate using task-specific metrics described in detail in Appendix Section H.

**Temporal Scene Classification** TEOChat achieves impressive performance on TSC with the fMoW RGB (75.1%) and Sentinel (45.5%) validation sets, outperforming both Video-LLaVA (16.6%, 4.9%) and GeoChat (59.2%, 26.3%) (Table 1). We also compare TEOChat to linear probing a specialist encoder that was pre-trained with self-supervision on fMoW images (SatMAE). TEOChat obtains considerably higher accuracy than SatMAE on fMoW RGB (+9.2%), and emphasize the TEOChat is a generalist which is not pre-trained on EO imagery. We also note that, impressively, TEOChat performs comparably to SatMAE+Stack (-0.8%) and is approaching state-of-the-art (SOTA) specialist performance on fMoW RGB (-4.1%) (Khanna et al., 2024a).

**Change Detection and Spatial Referring Expression** TEOChat outperforms Video-LLaVA and GeoChat by a substantial margin on all CD tasks (Table 1). However, on xBD localization, it underperforms the specialist by a large margin. TEOChat's CD performance on S2Looking is higher than the specialist model's performance (+8.0 $F_1$). However, we note that SOTA performance on S2Looking CD (Li et al., 2024b) and xBD localization Chen et al. (2024a) is much higher than TEOChat (+31.3 $F_1$, +48.5 $F_1$), which we believe is largely because these bitemporal change detection models output segmentation masks, whereas TEOChat is constrained to output boxes. Qualitatively, TEOChat does moderately well at identifying building locations but struggles to precisely localize boundaries and localize large numbers of buildings. This may be partially due to representing

| LLM/Projector Initialization | Image Encoder Initialization | Projector | Image References | Fine-tuned on TEOChatlas | Train Iters | fMoW RGB | xBD Dmg Cls. | S2Looking Det. | QFabric RQA-2 | QFabric RQA-5 |
|---|---|---|---|---|---|---|---|---|---|---|
| Video-LLaVA | CLIP | Frozen | - | - | 7k | 16.6 | 8.3 | 6.2 | 22.6 | 26.5 |
| Video-LLaVA | CLIP | Frozen | - | ✓ | 7k | 63.2 | 22.2 | 9.9 | 59.0 | 65.7 |
| LLaVA | CLIP | Frozen | ✓ | ✓ | 7k | 66.8 | 8.9 | 4.8 | 58.7 | 64.6 |
| Video-LLaVA | SkyScript | Frozen | ✓ | ✓ | 7k | 55.6 | 3.7 | 4.0 | 29.9 | 50.0 |
| Video-LLaVA | SkyScript | Fine-tuned | ✓ | ✓ | 7k | 68.3 | 40.8 | 12.5 | 59.0 | 66.6 |
| Video-LLaVA | CLIP | Frozen | ✓ | ✓ | 7k | 69.2 | 40.6 | 18.1 | 61.4 | 67.9 |
| Video-LLaVA | CLIP | Fine-tuned | ✓ | ✓ | 7k | 71.7 | 47.2 | 22.7 | 61.7 | 69.2 |
| Video-LLaVA | CLIP | Frozen | ✓ | ✓ | 14k | 71.5 | 45.8 | 30.4 | 66.6 | 74.3 |
| Video-LLaVA | CLIP | Fine-tuned | ✓ | ✓ | 14k | **75.1** | **50.0** | **33.6** | **66.7** | **74.3** |

Table 2: **Model ablations testing the impact of different design decisions in TEOChat.** Our design (bottom row) outperforms all other tested designs on the five canonical temporal tasks.

object locations using natural language tokens, and partially due to using axis-aligned boxes rather than general polygons. On damage classification, TEOChat outperforms the specialist substantially (+23.5 $F_1$). On SRE, TEOChat achieves much higher performance than Video-LLaVA (+21.4 $F_1$ on xBD, +31.2 $F_1$ on S2Looking) and GeoChat (+16.4 $F_1$ on xBD, +28.4 $F_1$ on S2Looking).

**Change Question Answering and Region-Based Change Question Answering** On QA and RQA, TEOChat achieves much higher performance than the baseline VLMs on all tasks (Table 1). On QA, TEOChat achieves 89.9% and 73.4% accuracy on xBD and S2Looking respectively, compared to GeoChat which attains 34.0% and 51.5% on the respective datasets. Similarly, on RQA, TEOChat outperforms GeoChat by 30.4% accuracy on xBD and 22.8% on S2Looking, and drastically outperforms on both QFabric tasks. TEOChat underperforms both specialist models on QFabric but achieves a much more similar $F_1$ to the specialists compared to both GeoChat and Video-LLaVA.

**Temporal Referring Expression and Region-based Temporal Question Answering** We find TEOChat obtains much higher accuracy than Video-LLaVA on TRE (74.9% vs. 1.9%) and RTQA (71.7% vs. 26.5%). On 5 image RTQA in QFabric, TEOChat outperforms the specialist by 11.9 $F_1$.

## 5.2 MODEL ABLATIONS

**Temporal vs. joint single-temporal training.** We investigate the effect of joint single image and temporal training by training a model on only the temporal tasks (TEOChat-T). TEOChat-T underperforms TEOChat on all but two tasks, matching it on fMoW Sentinel and slightly outperforming it on S2Looking QA (Table 1). This suggests joint training also largely benefits temporal capabilities.

**Model design** We test various design decisions of TEOChat. Specifically, we measure the impact of using a LLaVA initialization instead of Video-LLaVA, using an image encoder initialization from strong remote sensing pre-trained weights (SkyScript (Wang et al., 2024d)) instead of CLIP, fine-tuning versus freezing the projector layer, and including image references in the prompt. We note that we couple the LLM and projector initialization as they were pretrained together. Furthermore, to fairly compare to a new image encoder initialization, we also test the impact of using a fine-tuned projector with the alternate initialization to allow it to adapt to the new image encoder outputs. Finally, we measure the performance improvement from training on TEOChatlas (without using image references) and the effect of training for longer (7k iters [1 epoch] vs 14k [2 epochs]).

Our design of TEOChat outperforms all tested designs on the five canonical temporal tasks (Table 2). Fine-tuning on TEOChatlas leads to large performance improvements across the tasks and including image references leads to further improvements. Interestingly, image references help even in the bitemporal case, which is likely because the 'Image 2' reference helps delineate the visual tokens between the two images, as it explains most of the improvement from including references on the canonical bitemporal tasks (Table 9). The benefits of image references hold at test-time as well, as their exclusion from the test-time prompt leads to substantial performance drops on both tasks (Table 10). Using LLaVA weights for the LLM and projector initialization underperforms VideoLLaVA across all tasks which we attribute to the fact that Video-LLaVA is a stronger model and was trained with natural video data. The use of remote sensing pre-trained image encoder also underperforms a CLIP-pretrained image encoder even with projector fine-tuning, which we believe is partially because the LLM weights were pre-trained with a frozen CLIP-initialized image encoder during multi-modal instruction-tuning on natural video. We also find that fine-tuning outperforms freezing the projector, and training the model for longer leads to further performance improvements on all tasks, with especially large improvements on fMoW RGB and xBD damage classification.

| Model | ABCD | CDVQA |
|---|---|---|
| Specialist | 95.2 | 63.4 |
| Video-LLaVA | 50.0 | 29.8 |
| GeoChat | - | - |
| TEOChat | **85.6** | **47.2** |

| Model | xBD Dmg Cls. | S2Looking Det. |
|---|---|---|
| InternVL2 (Chen et al., 2024b) | 11.8 | 7.8 |
| Qwen2VL (Wang et al., 2024b) | 17.3 | 9.9 |
| Gemini 1.5 Pro | 35.8 | 16.5 |
| GPT-4o | 38.3 | 21.5 |
| TEOChat | **50.0** | **33.6** |

Table 3: **Zero-shot ABCD (building CD) and CDVQA (change QA) results.**

Table 4: **Comparison to other multi-image VLMs.** GPT-4o and Gemini 1.5 Pro use three in-context examples.

| Model | AID | UCMerced | LRBEN | | | HRBEN | | Average |
|---|---|---|---|---|---|---|---|---|
| | | | Pres. | Comp. | Rural/Urban | Pres. | Comp. | |
| Video-LLaVA (Lin et al., 2023) | 52.4 | 46.5 | 55.8 | 65.1 | 61.0 | 64.5 | 66.8 | 58.5 |
| RSVQA (Lobry et al., 2020) | - | - | *87.5* | *81.5* | *90.0* | *90.4* | *88.2* | - |
| RSGPT (Hu et al., 2023) | - | - | *91.2* | *91.7* | *94.0* | *90.9* | *90.0* | - |
| LHRS-Bot (Muhtar et al., 2024) | 91.3 | - | *88.5* | *90.0* | *89.1* | *92.6* | *92.5* | - |
| VHM (Pang et al., 2024) | **91.7** | - | *91.1* | *92.0* | ***95.0*** | 64.0 | **83.5** | - |
| GeoChat (Kuckreja et al., 2023) | 72.0 | 84.4 | *91.1* | *90.3* | *94.0* | 58.5 | 83.2 | 79.9 |
| TEOChat (Ours) | 80.9 | **86.3** | ***91.7*** | ***92.7*** | *94.0* | **67.5** | 81.1 | **83.4** |

Table 5: **Single image accuracies for scene classification (AID, UCMerced) and VQA (LRBEN, HRBEN).** We *italicize* results from fine-tuning on the corresponding training set. All other results are zero-shot. We **bold** the highest zero-shot score, except LRBEN which all models but Video-LLaVA are fine-tuned on.

## 5.3 Zero-shot performance on change detection tasks

We assess TEOChat on two new temporal remote sensing datasets, namely change detection using ABCD (Fujita et al., 2017) and change visual question answering using CDVQA (Yuan et al., 2022) (Table 3). TEOChat demonstrates impressive generalization to these datasets, outperforming Video-LLaVA (+35.6 on ABCD, +17.4 on CDVQA) and approaching specialist performance.

## 5.4 Comparison to other multi-image VLMs

We compare TEOChat to open-source multi-image VLMs InternVL2-4B (Chen et al., 2024b) and Qwen2VL-7B-Instruct (Wang et al., 2024b) as well as GPT-4o and Gemini 1.5 Pro with three demonstrating examples on xBD damage classification and S2Looking change detection (Table 4). We focus this analysis on these datasets as we could not evaluate the proprietary models on fMoW or QFabric (larger both in size and sequence length) due to cost. TEOChat outperforms both Qwen2-VL and InternVL2 by a large margin on xBD damage classification (+38.2 F1, +32.7 F1 respectively) and S2Looking change detection (+25.8 F1, +23.7 F1 respectively). TEOChat also performs better than both proprietary models on xBD damage classification (+11.7 F1 compared to GPT-4o, +12.8 F1 compared to Gemini 1.5 Pro) and S2Looking change detection (+14.2 F1 compared to GPT-4o, +17.1 F1 compared to Gemini 1.5 Pro). Qualitatively, the proprietary models tend to under-predict change in S2Looking and underestimate building damage in xBD, whereas the open-source models produce highly inaccurate boxes and classify almost all examples as 'No Damage'.

## 5.5 Single image performance

We measure TEOChat's single EO image capabilities by evaluating it on (1) scene classification tasks (AID (Xia et al., 2017) and UCMerced (Yang & Newsam, 2010)) and (2) visual question answering tasks (LRBEN and HRBEN (Lobry et al., 2020)). We note that evaluation on AID, UCMerced, and HRBEN is zero-shot, as they are not in TEOChat's training set.

TEOChat matches or exceeds the performance of all single image models on six of the seven tasks (Table 5). It outperforms GeoChat on both LULC classification tasks, notably by a large margin on AID (+8.9), a much larger-scale dataset than UCMerced. It underperforms LHRS-Bot and VHM on AID, likely due to the inclusion of several more LULC classification datasets in their training data. TEOChat also outperforms all models on low resolution VQA presence and comparison tasks, and attains close to the best performance on rural/urban classification. It achieves the highest zero-shot performance on high resolution (HR) presence, while slightly underperforming GeoChat and VHM on HR comparison. Notably, GeoChat and VHM use higher resolution images than TEOChat (504 and 336 vs. 224). TEOChat attains higher average performance than GeoChat by a considerable

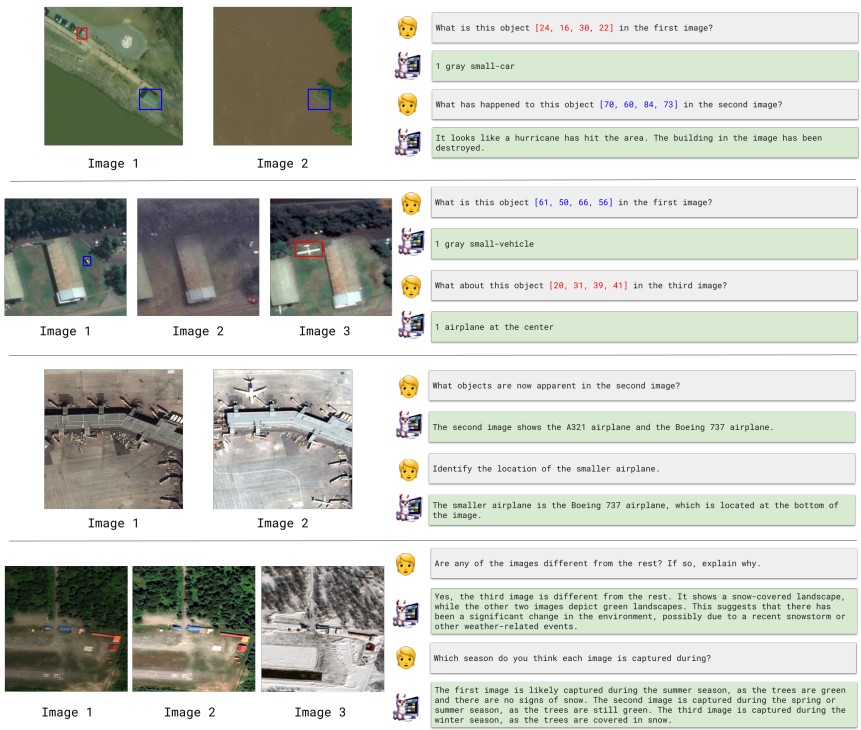

Figure 4: **Qualitative examples of task generalization exhibited by TEOChat on temporal EO images.**

margin (+3.5), suggesting it has stronger single image capabilities on top of the new temporal ones. Finally, we measure TEOChat's performance on three EO caption datasets. We find it outperforms GeoChat on all datasets and matches other single image EO generalists on the larger two datasets in the fine-tuning setting (Tables 11 to 14), and matches or outperforms GeoChat on average in the zero-shot setting (Tables 15 to 18). Figures 7 and 8 show example TEOChat single image responses.

### 5.6 TEOCHAT GENERALIZATION

TEOChat also exhibits interesting cases of generalization to tasks not explicitly represented in the training data (Figure 4). For example, TEOChat can compose its spatial and temporal reasoning abilities, such as identifying objects that are only regionally annotated in the single image training tasks (like airplanes and cars) within different images in the temporal sequence. It can also recognize differences between images (like seasonality) which is not explicitly reflected in the training data.

## 6 CONCLUSION

In this work, we propose TEOChat, the first vision and language assistant that can engage in conversation about temporal EO imagery. We introduce a novel dataset called TEOChatlas to train TEOChat on a diverse set of spatial and temporal instruction-following tasks. TEOChat exhibits strong performance, outperforming previous VLMs substantially, and even competing with or outperforming several specialist models. TEOChat also demonstrates impressive zero-shot generalization to two change detection datasets and outperforms strong multi-image proprietary models .

There are several interesting directions for future work. First, it would be interesting to explore other architectures to process the visual data across time. The LLM is responsible for performing the temporal information aggregation in TEOChat, but it may be helpful to use a dedicated part of the architecture for this. Second, it would be useful to enable the model to process multi-spectral bands which is common in EO data. Third, exploring methods to improve object localization is worthwhile, including regressing the coordinates instead of representing them with natural language tokens (Zhang et al., 2023a), using oriented bounding boxes to allow for tighter localization, or increasing memory efficiency of the architecture to enable inputting higher resolution images.

# 7 REPRODUCIBILITY STATEMENT

Training data, model weights, training code, and evaluation code are hosted at `https://github.com/ermongroup/TEOChat`. All major experimental results can be reproduced by following the steps described there.

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

APPENDIX

A  RELATED WORK

**ML for Temporal Earth Observation** ML methods have been commonly used to model temporal sequences of earth observation data (Moskolaï et al., 2021; Miller et al., 2024; Khanna et al., 2024b). Change detection is a widely studied earth observation task where the goal is to identify change between pairs (bitemporal) or sequences (multitemporal) of images capturing the same location over time (Chughtai et al., 2021; Bai et al., 2023; Cheng et al., 2024). Automated methods for change detection can aid important humanitarian and sustainability efforts, including planning disaster relief (Rahnemoonfar et al., 2021; Gupta et al., 2019; Ban et al., 2020), tracking urban changes (Tamilenthi & Baskaran, 2013; Fyleris et al., 2022; Verma et al., 2021), and monitoring ecological conditions (Willis, 2015; Xu et al., 2019), deforestation (Shermeyer & Haack, 2015; De Bem et al., 2020; Decuyper et al., 2022), and crops (Gim et al., 2020; Kaur et al., 2023; Wang et al., 2024a). Commonly studied change detection ML subtasks include building change detection (Shen et al., 2021) and damage detection (Gupta et al., 2019), semantic change detection including land cover changes (Daudt et al., 2019; Yang et al., 2021; Toker et al., 2022) and land use changes (Verma et al., 2021). Many ML-based approaches have been developed to accomplish these tasks, and almost all use a Siamese architecture where the vision encoder is shared across time (Zheng et al., 2021d;b; 2022b; Liu et al., 2024c; Tian et al., 2023), which we adopt as well.
Beyond change detection, multitemporal scene classification using the Functional Map of the World (fMoW) is a common benchmark (Christie et al., 2018). Finally, while several works have designed self-supervised approaches to leverage temporal sequences of earth observation data (Ayush et al., 2021; Manas et al., 2021; Cong et al., 2022), none have developed vision-language models that can perform a wide variety of instruction-following tasks on earth observation imagery over time.

**VLMs for Natural Images** There have been substantial recent advancements in the development of vision-language models (VLMs) for natural images, notably including LLaVA (Liu et al., 2024a) and its successor LLaVA-1.5 (Liu et al., 2023a). These seminal works have demonstrated the effectiveness of curating large multimodal instruction-following datasets to train high performing vision-language models that can respond to user instructions about image data using a simple model design consisting of a vision encoder to get visual representations, a projector layer to convert visual tokens into language space, and an LLM to decode the instructions and converted visual tokens into a response. Subsequent works have proposed approaches to build upon these instruction-tuned VLMs, from improving performance with mixture of experts (Lin et al., 2024), achieving similar performance with fewer parameters (Zhou et al., 2024; Zhu et al., 2024), and improve task transfer (Li et al., 2024a) to adding new capabilities like object detection (Wang et al., 2024c), semantic segmentation (Rasheed et al., 2023; Zhang et al., 2023a), image generation (Sun et al., 2023), and tool use (Liu et al., 2023b).

**Temporal VLMs** Several recent works have developed VLMs that can engage in conversation about videos. VideoChatGPT curates a video instruction set to train a LLaVA-style architecture but combining both spatially pooled temporal features and temporally pooled spatial features on CLIP-encoded frames (Maaz et al., 2023). VideoLLaVA then demonstrates improved performance by training a LLaVA-style architecture with separate, pre-aligned image and video encoders on a joint instruction-following dataset of both images and videos (Lin et al., 2023). Subsequent works have primarily adopted LLaVA-style architectures as well, but include additional layers aimed to process visual tokens across time before inputting them into the LLM (Jin et al., 2023; Li et al., 2023; Zhang et al., 2023b; Liu et al., 2024b).

**VLMs for EO** Efforts to develop instruction-following VLMs for EO data have been rapidly increasing (Hu et al., 2023; Kuckreja et al., 2023; Muhtar et al., 2024; Pang et al., 2024). These methods commonly curate many different single image EO datasets and convert them to instruction-following tasks, then instruction-tune a LLaVA-like model on the curated dataset. Importantly, no prior work has developed VLMs for temporal sequences for EO except for CDChat (Noman et al., 2024), a concurrent work that fine-tunes LLaVA-1.5 on two bitemporal change detection datasets. In contrast, our work builds upon the more recent VideoLLaVA architecture, enabling the handling of sequences with up to eight images. Additionally, we develop a public instruction-following dataset including many different single image tasks as well as temporal tasks from four EO datasets, broadening the scope and capabilities of the model.

| Dataset | Examples (Images) | Temporal | Detection |
|---|---|---|---|
| GeoChat_Instruct (Kuckreja et al., 2023) | 309k | - | ✓ |
| SkyEyeGPT (Zhan et al., 2024) | 968k | - | ✓ |
| MMRS1M (Zhang et al., 2024) | 1M | - | ✓ |
| LHRS-Instruct (Muhtar et al., 2024) | 42k | - | ✓ |
| ChatEarthNet (Yuan et al., 2024) | 173k | - | - |
| H2RSVLM-Instruct (Pang et al., 2024) | 180k | - | ✓ |
| SkySenseGPT (Luo et al., 2024) | 1.4M | - | ✓ |
| RS-GPT4V (Xu et al., 2024) | 991k | - | ✓ |
| TEOChatlas (Ours) | 554k (1.2M) | ✓ | ✓ |

Table 6: **Comparison of TEOChatlas to other instruction-following EO datasets.** TEOChatlas has a large number of instruction-following examples, and is the first EO dataset to have temporal tasks.

| Dataset | Temporal | Task | Sensor | # Examples |
|---|---|---|---|---|
| GeoChat | Single | Object Detection, VQA Scene Classification | GaoFen, Jilin-1, Sentinel-2, DJI Mavic Pro | 308,861 |
| fMoW | Multitemporal | Sequence Classification | WorldView-2/3, Sentinel-2 | 83,412 |
| xBD | Bitemporal | Building Damage Detection | WorldView-2 | 19,749 |
| S2Looking | Bitemporal | Building Change Detection | GaoFen, SuperView, BeiJing-2 | 17,090 |
| QFabric | Multitemporal | Multi-Task Urban Change Detection | WorldView-2 | 124,959 |

Table 7: **Datasets included in the TEOChatlas training set**. TEOChatlas is composed of single image and temporal image datasets from a variety of sensors and includes multiple instruction-following tasks for both spatial and temporal reasoning.

## B    TRAINING DATASETS

TEOChatlas consists of GeoChat_Instruct (Kuckreja et al., 2023), an instruction-following dataset with single earth observation images, as well as four temporal earth observation benchmarks (fMoW (Christie et al., 2018), xBD (Gupta et al., 2019), S2Looking (Shen et al., 2021), and QFabric (Verma et al., 2021)). We describe how we process each dataset in more detail below and a summary of the dataset statistics is provided in Table 7. We note that for all datasets, we (1) resize the shorter side to 224 then crop to 224x224 before inputting the images into the network and (2) randomly sample between all instruction-following tasks when creating examples, sometimes randomly sampling between different phrasings of the same task to introduce diversity in the wording. A spatial map of the locations of the examples in TEOChatlas is shown in Figure 5 and a comparison of TEOChatlas to other instruction-following EO datasets is shown in Table 6.

**GeoChat_Instruct** The GeoChat_Instruct dataset (Kuckreja et al., 2023) is an instruction-following dataset composed of multiple single image EO datasets spanning tasks including object detection derived from SAMRS (which is itself a composition of DOTA (Xia et al., 2018), DIOR (Li et al., 2020), and FAIR1M (Sun et al., 2022)), visual question answering from LRBEN (Lobry et al., 2020) and Floodnet (Rahnemoonfar et al., 2021), and scene classification from NWPU-RESISC-45 (Cheng et al., 2017). The instruction-following tasks in the GeoChat_Instruct dataset include scene classification (classify the image into one of several classes provided in the prompt), visual question answering (answer a question about the image), referring expression (identify the location of an object given an expression describing its characteristics), region captioning (describe a location represented as a bounding box in the prompt), detailed description (describing the whole image in detail), grounded description (describing the whole image in detail with bounding boxes included in the description), and multi-round conversation (multiple turns of conversation). In total, the GeoChat_Instruct training dataset consists of 308,861 single instruction-following examples on 106,747 images.

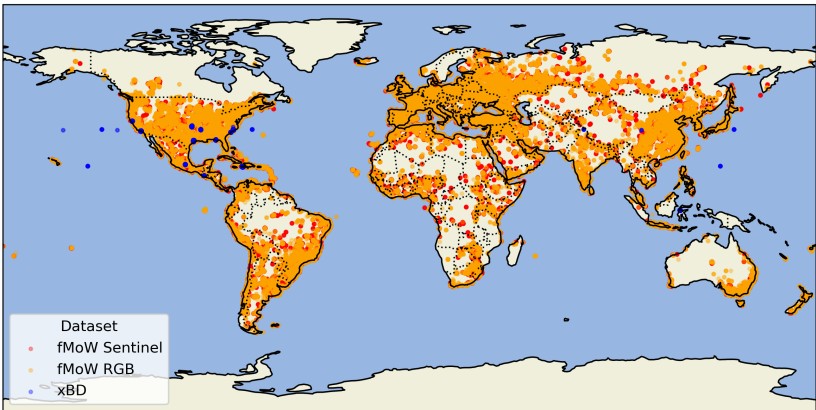

Figure 5: **Locations of the examples in the xBD and fMoW RGB, and fMoW Sentinel subsets of the TEOChatlas dataset.** GeoChat, S2Looking, and QFabric do not have geographic locations in the released data. However, the datasets in GeoChat span many locations around the world, and both S2Looking and QFabric contain globally distributed locations as well.

**fMoW** We use both the fMoW RGB dataset (Christie et al., 2018) and the fMoW Sentinel dataset (Cong et al., 2022), sampling randomly between the two for each example in the provided training set split. We follow standard practice for converting fMoW to a classification dataset and crop each image to the provided bounding box. We create one instruction-following example for every image in the training set, leading to 83,412 examples on 83,412 images. We report model performance on both the fMoW RGB and fMoW Sentinel validation sets.

**xBD** We include the xBD (Gupta et al., 2019) training dataset in TEOChatlas. To both increase the representation of building change detection in TEOChatlas and maintain image resolution, we crop the 1024x1024 images into 16 256x256 images. When creating instruction-following tasks, we convert all building polygons to their minimal axis-aligned bounding boxes. We create one instruction-following example for every image in the training set, leading to 19,749 examples on 19,749 images. We use the original polygons when creating the segmentation masks for evaluation (see below), and report model performance on the test set.

**S2Looking** We include the S2Looking (Shen et al., 2021) training dataset in TEOChatlas. Similarly to xBD, we crop the 1024x1024 images into 16 256x256 images and convert all building polygons to their minimal axis-aligned bounding boxes. We create one instruction-following example for every image in the training set, leading to 17,090 examples on 17,090 images. We use the original polygons when creating the segmentation masks for evaluation (see below), and report model performance on the test set.

**QFabric** We include the QFabric (Verma et al., 2021) training dataset in TEOChatlas. The images in QFabric cover a large area with 10,000 x 10,000 pixels, which we crop into around 1600 256x256 images. We drop any cropped images that have no polygons and convert all polygons to minimal axis-aligned bounding boxes. We create one instruction-following example for every image in the training set, leading to 124,959 examples on 124,959 images. We use the original polygons when creating the segmentation masks for evaluation (see below), and report performance on the test set.

## C  TEMPORAL INSTRUCTION-FOLLOWING TASKS

We provide a summary of all temporal instruction-following tasks included in the TEOChatlas training set in Table 8. For description and captioning temporal tasks, we use GPT-4o to increase the diversity of the responses to avoid template-fitting which we found plagued the model in early experiments.

| Dataset | Task Category | Example |
|---|---|---|
| fMoW | Sequence Classification | **User:** What class does this sequence of images belong to?
**AI:** Lake or pond. |
| xBD | Localization | **User:** Identify all the buildings in the image.
**AI:** \<bbox\>, \<bbox\>, \<bbox\>, \<bbox\>, \<bbox\>. |
| | Classification | **User:** Classify the level of damage experienced by the building at location \<bbox\> in the second image. Choose from: No damage, Minor Damage, Major Damage, Destroyed.
**AI:** Minor Damage. |
| | Change Question Answering | **User:** What disaster has occurred here?
**AI:** An earthquake.
**User:** Which part of the image was most affected by the disaster?
**AI:** The center left of the image was most affected by the disaster.
**User:** How many buildings in the image have been destroyed?
**AI:** 3.
**User:** Are there any damaged buildings in the image?
**AI:** Yes. |
| | Region-based Question Answering | **User:** Are there any damaged buildings in this region \<bbox\>?
**AI:** Yes.
**User:** How severe is the damage to this building \<bbox\>?
**AI:** The given building has been destroyed. |
| | Spatial Change Referring Expression | **User:** Identify the severely damaged or destroyed buildings in the image.
**AI:** \<bbox\>, \<bbox\>.
**User:** Identify the damaged buildings in the ¡section¿ of the image.
**AI:** There are no damaged buildings in the bottom right of the image.
**User:** Identify the destroyed buildings in this area \<bbox\>.
**AI:** There are no destroyed buildings in the given area. |
| | Region Captioning | **User:** How has this building \<bbox\> changed?
**AI:** The given building has been destroyed. |
| | Detailed Description | **User:** Describe how the buildings have changed.
**AI:** There has been a flood that has damaged multiple buildings in the area. |
| | Grounded Description | **User:** Describe how the buildings have changed. Include bounding boxes.
**AI:** There has been a flood that has damaged a building at the location \<bbox\>. |
| S2Looking | Localization | **User:** Identify all changed buildings.
**AI:** \<bbox\>, \<bbox\>, \<bbox\> |
| | Change Question Answering | **User:** Have any buildings been constructed in the area? Please answer with Yes or No.
**AI:** Yes.
**User:** How many buildings in the image have been built or destroyed?
**AI:** 3. |
| | Region-based Temporal Question Answering | **User:** Has the area \<bbox\> changed? Please answer with Yes or No.
**AI:** Yes. |
| | Spatial Change Referring Expression | **User:** Identify the {constructed, destructed} buildings in the image.
**AI:** \<bbox\>, \<bbox\>, \<bbox\>
**User:** What is the largest building that experienced a change?
**AI:** \<bbox\>. |
| | Detailed Description | **User:** Provide a detailed description of the buildings that have changed.
**AI:** Five buildings have been constructed. |
| | Grounded Description | **User:** Provide a detailed description of the buildings that have changed. Include bounding boxes in your output.
**AI:** Three buildings have been destroyed \<bbox\>, \<bbox\>, \<bbox\>. |
| | Region Captioning | **User:** Describe how the buildings have changed in this area: \<bbox\>.
**AI:** Five buildings have been constructed. |
| QFabric | Temporal Referring Expression | **User:** Identify all images in which land was cleared in this region \<bbox\> from the previous image.
**AI:** Image 1, Image 2
**User:** In which images is \<class\> visible in this region \<bbox\>?
**AI:** Image 1, Image 2, Image 3
**User:** Identify all images in which construction occurred in this region \<bbox\> from the previous image.
**AI:** Image 4
**User:** Identify all images in which a construction project was begun in this region, area \<bbox\> from the previous image.
**AI:** Image 3
**User:** Identify the image when \<change phrase\> in this region \<bbox\>.
**AI:** Image 2 |
| | Region-Based Question Answering | **User:** Has there been urban development in this area \<bbox\>?
**AI:** Yes
**User:** Identify the type of urban development that has occurred in this area \<bbox\>.
**AI:** Residential |
| | Temporal Region-Based Question Answering | **User:** Has there been urban development in this area \<bbox\> between image N and image M?
**AI:** No
**User:** What is the development status in this region \<bbox\> in image N?
**AI:** Finished Construction |
| | Region Captioning | **User:** How has the area \<bbox\> changed as a result of urban development?
**AI:** This region was greenland at first, and then became finished construction. |
| | Temporal Region Captioning | **User:** How has the area \<bbox\> changed due to urban development from image N to image M?
**AI:** Between the first and fourth images, this area was cleared land at first, and then became midway-finished construction and then became finished construction. |

Table 8: **Examples of the many temporal instruction-following tasks in TEOChatlas.**

## D  SINGLE IMAGE DATASETS

We evaluate TEOChat on the following datasets:

1. AID (Xia et al., 2017), a scene classification dataset composed of 10,000 images each classified into one of 30 scene classes.

2. UCMerced (Yang & Newsam, 2010), a land use classification dataset consisting of 2100 images each classified into one of 21 land use classes.

3. LRBEN and HRBEN (Lobry et al., 2020), low resolution and high resolution datasets for visual question answering. We consider the presence task (answering yes or no for whether an object is in the image) and comparison tasks (answering yes or no to a comparison question, for example are there more buildings than roads in the image).

## E    TEOCHAT PROMPTS

We use the following system prompt for the tasks in TEOChatlas.

> A chat between a curious user and an artificial intelligence assistant. The assistant gives helpful, detailed, and polite answers to the user's questions. USER: This is a sequence of {high resolution, optical} satellite images {from <sensor>}: <video>.

where we randomly inject the information in curly brackets when metadata (resolution or sensor) is available and replace the "<video>" token with the "<image>" token $N$ times, with $N$ the number of images in the sequence. After this prompt we add the task instruction. Then, for tasks that require a choice of options, we provide the options in the prompt. For tasks that require bounding boxes to be output, we include "Please include bounding boxes of the form [x_min, y_min, x_max, y_max] in your response." While this does not guarantee correct output parsing in every case, fewer than 0.1% errors occurred across all evaluations we conducted. Finally, we add "ASSISTANT:".

## F    ADDITIONAL TRAINING DETAILS

We fine-tune the LLM in TEOChat using LoRA rank 128. Before inputting the images into the image encoder, we resize the shorter dimension of each image to 224 pixels, and then apply a center crop to obtain a 224x224 image. These design decisions allow us to train the model with sequences of up to 8 images on a single NVIDIA A4000 GPU (16B of VRAM) with a batch size of 1. To increase the effective batch size, we use 8 steps of gradient accumulation and train the network on 10 GPUs using data parallelism together with optimizer state partitioning, gradient partitioning, parameter partitioning, and CPU offloading with DeepSpeed Zero-3 Offload (Rasley et al., 2020). We optimize the network using AdamW (Loshchilov & Hutter, 2017) with a cosine learning rate scheduler, a peak learning rate of 2e-5, and a warmup of 3% of the epoch. The model takes 125 hours to train per epoch and we train for 2 epochs.

## G    GEOCHAT TEMPORAL POST-PROCESSING

We describe how we obtain GeoChat predictions on the temporal prediction tasks below.

### G.1    TEMPORAL SCENE CLASSIFICATION

We run GeoChat on each image in the sequence to produce a per-image classification, then take the majority vote across the sequence as GeoChat's prediction over the whole sequence.

### G.2    CHANGE DETECTION

**xBD Building Localization**  We prompt GeoChat to identify all buildings in the first image then use the output bounding boxes to obtain the predicted segmentation mask.

**xBD Damage Classification**  We input each ground truth box and the second image into GeoChat as separate examples, instructing it to classify the given box into the four damage categories, and use the classification labels output by GeoChat with the boxes to construct the segmentation mask.

**S2Looking Building Change Detection** We prompt GeoChat to identify all buildings in first and second images individually. From the bounding box predictions on each image, we create binary segmentation masks then take the difference to obtain GeoChat's predicted change mask, masking out pixels from overlapping boxes across the two images to avoid small deviations in box predictions between the two images from harming performance.

### G.3 SPATIAL REFERRING EXPRESSION

**xBD** We use the same procedure as for building localization, but prompt GeoChat to identify only the 'destroyed' buildings, for example.

the spat**S2Looking** The spatial referring expression tasks for S2Looking are to identify the constructed or destructed buildings. For both, we prompt GeoChat to identify all buildings in the first image and in the second image, construct a segmentation mask for each, then take a difference. We use the pixels in the first image but not the second as destructed and the inverse as constructed.

### G.4 CHANGE QUESTION ANSWERING

**xBD** As the xBD questions pertain to building damage levels and there is no damage to buildings in the pre-disaster images, we input the second image and the question (e.g. are there any severely damaged or destroyed buildings in this area?) to GeoChat to obtain the answer.

**S2Looking** The change questions for S2Looking ask the model to identify whether or not buildings changed between the images. To obtain this binary prediction from GeoChat, we follow a similar approach and instruct it to identify the buildings in the first image and the buildings in the second, then consider the predicted building from each image as changed if it does not overlap a box in the other image. If any buildings changed, the binary prediction is yes, otherwise it is no.

### G.5 REGION-BASED CHANGE QUESTION ANSWERING

**xBD** As the input bounding box is the only difference between this task and change question answering, we use similar postprocessing as that task, inputting the second image and a question as well as a bounding box to GeoChat to obtain the answer.

**S2Looking** Similarly, this task only differs from change question answering due to the bounding box input, so we perform the same postprocessing as for that task, but only taking into account the changes inside the input bounding box.

**QFabric** These tasks are standard classification tasks, so we follow a similar procedure as for temporal scene classification and feed each image in the sequence (either 2 or 5 images) along with the bounding box in the prompt into GeoChat, then take a majority vote to get the final classification.

### H EVALUATION SETUP

For previously existing tasks, we follow the metrics reported in the original works. For new tasks, we use accuracy. See further detail below.

**Change Detection and Spatial Referring Expression** To evaluate TEOChat on building change detection (CD) and spatial referring expression (SRE), we follow the standard performance metrics of the building localization (Loc.) and damage classification (Dmg Cls.) subtasks on xBD and building change detection (Det.) subtask on S2Looking. Specifically, for building localization and building change detection, we convert the bounding box predictions and ground truth polygons to segmentation masks and compute per-pixel $F_1$ between the predicted and ground truth segmentation masks. For building damage classification, we input the ground truth polygon to the model to obtain a classification label, then create a segmentation mask where the pixels within the ground truth polygon are assigned the predicted label, and compute class-weighted $F_1$ between this mask and the ground truth mask. For SRE, we use the same evaluation scheme as building localization.

**Change Question Answering and Region-Based Change Question Answering** We measure accuracy on change question answering (QA) and region-based change question answering (RQA) for xBD and S2Looking. For QFabric, we use the proposed evaluation metrics from Verma et al. (2021) for comparability to the specialists, namely per-pixel $F_1$ after converting the classified polygons to

segmentation masks as also done in xBD damage classification. We consider both a 2 image task and 5 image task as done in Verma et al. (2021).

**Temporal Referring Expression and Region-based Temporal Question Answering** For temporal referring expression (TRE), we measure performance of the models to identify the correct image(s) using accuracy. For region-based temporal question answering (RTQA), we use F1 for the change status task (classifying the development status of each image in a sequence of 5 images) following Verma et al. (2021), and use accuracy for the other tasks.

## I ADDITIONAL RESULTS

### I.1 IMAGE REFERENCE ABLATIONS

We run a ablation to investigate why image identifiers improve performance on the canonical bitemporal tasks (Table 9) and run another ablation to measure the importance of including the image identifiers in the prompt on the temporal referring tasks (Table 10).

| Image Reference Strategy | xBD Dmg Cls. ($F_1$) | S2Looking Det. ($F_1$) |
|---|---|---|
| No References | 44.7 | 25.5 |
| Only 'Image 1' | 47.5 | 25.8 |
| Only 'Image 2' | 49.2 | 34.5 |
| Both 'Image 1' and 'Image 2' | 50.0 | 34.5 |

Table 9: **Impact of different image reference strategies on the canonical bitemporal datasets.** Including 'Image 2' leads to much larger performance gains than including 'Image 1', constituting most of the performance improvement from including both image references.

| Image Identifiers | RTQA QFabric (Acc.) | TRE QFabric (Acc.) |
|---|---|---|
| - | 59.2 | 58.6 |
| ✓ | 71.7 | 74.9 |

Table 10: **Impact of including image references in the test-time prompt on TEOChat's performance on the region-based temporal question answering and temporal referring expression tasks.** Including image references leads to substantial benefits on both tasks.

### I.2 FINE-TUNED CAPTIONING RESULTS

We evaluate TEOChat on three EO captioning datasets used as benchmarks in prior work (Hu et al., 2023; Zhan et al., 2024), namely UCM-Captions (Qu et al., 2016), Sydney-Captions Qu et al. (2016), and RSICD (Lu et al., 2017). Given these works include these datasets in the training data, we follow Hu et al. (2023) and fine-tune TEOChat on each of the three datasets separately. We use the same training procedure as used while training on TEOChatlas except we fine-tune until convergence (5, 20, and 5 epochs respectively). For comparison we also fine-tune GeoChat as results on these datasets are not reported in Kuckreja et al. (2023), and use their same training procedure when fine-tuning on each dataset for the same number of epochs as TEOChat. We evaluate using the same metrics reported in Hu et al. (2023) and Zhan et al. (2024). Fine-tuning results are reported in Tables 11 to 14 and zero-shot results are reported in Tables 15 to 18.

In the fine-tuning setting, we find TEOChat exceeds the performance of specialists and matches generalist single image EO models on the larger two of the three captioning datasets (UCM-captions and RSICD), and shows strong captioning abilities qualitatively on all three datasets (Figure 8). On Sydney-captions, TEOChat underperforms RSGPT, but qualitatively we find TEOChat's captions are high quality. Importantly, TEOChat outperforms GeoChat across all metrics on all three captioning datasets except METEOR on UCM-captions, for which it slightly underperforms GeoChat (-0.4%). In the zero-shot setting, TEOChat performs similarly to or outperforms GeoChat on average across all metrics, with notable improvements on the largest, more difficult captioning dataset (RSICD). Interestingly, MiniGPT-v2 achieves impressive zero-shot performance on these datasets.

We believe the explanation for strong fine-tuning performance of RSGPT and MiniGPT-v2's strong zero-shot and fine-tuning performance is twofold. First, we largely attribute this to the inclusion of a higher proportion of captioning datasets in their training data, whereas GeoChat and TEOChat have more scene classification and question answering tasks. Second, these three captioning datasets have several known issues which make their use as benchmarks suboptimal, including limited vocabulary and very short descriptions, and sometimes even misspelled words and grammatical errors. We caution researchers about using these benchmarks without careful analysis. We finally emphasize that in addition to these single image capabilities, TEOChat has temporal reasoning capabilities that none of these other models have.

| Model | BLEU-1 | BLEU-2 | BLEU-3 | BLEU-4 | METEOR | ROGUE_L | CIDEr |
|---|---|---|---|---|---|---|---|
| *Specialists* | | | | | | | |
| SAA(Lu et al., 2019) | 79.6 | 74.0 | 69.1 | 64.8 | 38.6 | 69.4 | 294.5 |
| Post-processing (Hoxha et al., 2023) | 79.7 | 73.0 | 67.4 | 62.6 | 40.8 | 74.1 | 309.6 |
| *Generalists* | | | | | | | |
| MiniGPT-4 (Zhu et al., 2023b) | 30.9 | 27.6 | 22.2 | 18.1 | 33.4 | 41.4 | 0.0 |
| Shikra (Chen et al., 2023b) | 81.2 | 58.9 | 43.3 | 34.0 | 32.6 | 56.7 | 56.7 |
| MiniGPT-v2 (Chen et al., 2023a) | 81.1 | 60.3 | 45.1 | 36.2 | 32.4 | 56.6 | 60.7 |
| RSGPT (Hu et al., 2023) | **86.1** | **79.1** | 72.3 | 65.7 | 42.2 | 78.3 | **333.2** |
| GeoChat (Kuckreja et al., 2023) | 82.9 | 74.9 | 68.5 | 63.0 | **44.4** | 78.1 | 299.1 |
| TEOChat | 85.2 | 78.2 | **72.4** | **67.4** | 43.9 | **79.3** | 316.5 |

Table 11: **Captioning performance on the UCM-captions dataset.**

| Model | BLEU-1 | BLEU-2 | BLEU-3 | BLEU-4 | METEOR | ROGUE_L | CIDEr |
|---|---|---|---|---|---|---|---|
| *Specialists* | | | | | | | |
| SAA (Lu et al., 2019) | 68.8 | 60.7 | 52.9 | 45.4 | 30.5 | 58.2 | 170.5 |
| Post-processing (Hoxha et al., 2023) | 78.4 | 69.9 | 63.2 | 57.2 | 39.5 | 71.1 | 255.5 |
| *Generalists* | | | | | | | |
| MiniGPT-4 (Zhu et al., 2023b) | 29.5 | 25.9 | 20.3 | 16.4 | 32.0 | 42.7 | 0.1 |
| Shikra (Chen et al., 2023b) | 77.5 | 53.2 | 37.0 | 27.8 | 29.4 | 53.3 | 26.8 |
| MiniGPT-v2 (Chen et al., 2023a) | 77.4 | 55.8 | 40.6 | 32.3 | 29.9 | 52.1 | 33.8 |
| RSGPT (Hu et al., 2023) | **82.3** | **75.3** | **68.6** | **62.2** | **41.4** | **74.8** | **273.1** |
| GeoChat (Kuckreja et al., 2023) | 74.1 | 61.2 | 53.1 | 47.0 | 34.2 | 67.8 | 184.3 |
| TEOChat | 75.1 | 64.6 | 56.9 | 50.5 | 36.2 | 69.5 | 204.6 |

Table 12: **Captioning performance on the Sydney-captions dataset.**

| Model | BLEU-1 | BLEU-2 | BLEU-3 | BLEU-4 | METEOR | ROGUE_L | CIDEr |
|---|---|---|---|---|---|---|---|
| *Specialists* | | | | | | | |
| SAA (Lu et al., 2019) | 59.4 | 45.1 | 35.3 | 28.1 | 26.1 | 49.6 | **132.4** |
| Post-processing (Hoxha et al., 2023) | 62.9 | 46.0 | 35.7 | 28.7 | 25.3 | 47.3 | 75.6 |
| *Generalists* | | | | | | | |
| MiniGPT-4 (Zhu et al., 2023b) | 34.0 | 31.8 | 25.8 | 20.6 | **33.2** | 40.7 | 0.1 |
| Shikra (Chen et al., 2023b) | 82.6 | 62.5 | 45.2 | 34.6 | 30.3 | 53.6 | 19.9 |
| MiniGPT-v2 (Chen et al., 2023a) | **83.1** | **64.6** | **47.8** | **37.3** | 30.2 | 54.0 | 52.4 |
| RSGPT (Hu et al., 2023) | 70.3 | 54.2 | 44.0 | 36.8 | 30.1 | 53.3 | 102.9 |
| GeoChat (Kuckreja et al., 2023) | 69.2 | 51.4 | 40.1 | 32.6 | 28.0 | 54.4 | 95.2 |
| TEOChat | 70.3 | 52.4 | 41.2 | 33.7 | 28.9 | **55.7** | 97.2 |

Table 13: **Captioning performance on the RSICD captions dataset.**

| Model | BLEU-1 | BLEU-2 | BLEU-3 | BLEU-4 | METEOR | ROGUE_L | CIDEr |
|---|---|---|---|---|---|---|---|
| *Specialists* | | | | | | | |
| SAA (Lu et al., 2019) | 69.3 | 59.9 | 52.4 | 46.1 | 31.7 | 59.1 | 199.1 |
| Post-processing (Hoxha et al., 2023) | 73.7 | 63.0 | 55.4 | 49.5 | 35.2 | 64.2 | 213.6 |
| *Generalists* | | | | | | | |
| MiniGPT-4 (Zhu et al., 2023b) | 31.5 | 28.4 | 22.8 | 18.4 | 32.9 | 41.6 | 0.1 |
| Shikra (Chen et al., 2023b) | 80.4 | 58.2 | 41.8 | 32.1 | 30.8 | 54.5 | 34.5 |
| MiniGPT-v2 (Chen et al., 2023a) | **80.5** | 60.2 | 44.5 | 35.3 | 30.8 | 54.2 | 49.0 |
| RSGPT (Hu et al., 2023) | 79.6 | **69.5** | **61.6** | **54.9** | **37.9** | **68.8** | **236.4** |
| GeoChat (Kuckreja et al., 2023) | 75.4 | 62.5 | 53.9 | 47.5 | 35.5 | 66.8 | 192.9 |
| TEOChat | 76.9 | 65.1 | 56.8 | 50.5 | 36.3 | 68.2 | 206.1 |

Table 14: **Average performance across the three captioning datasets.**

| Model | BLEU-1 | BLEU-2 | BLEU-3 | BLEU-4 | METEOR | ROGUE_L | CIDEr |
|---|---|---|---|---|---|---|---|
| MiniGPT-v2 (Chen et al., 2023a) | 17.6 | 9.1 | 4.2 | 1.9 | **14.8** | **23.5** | **16.9** |
| GeoChat (Kuckreja et al., 2023) | **22.8** | **10.3** | **4.3** | 1.7 | 10.9 | 21.5 | 13.9 |
| TEOChat | 8.8 | 4.3 | 1.9 | 0.8 | 10.3 | 20.8 | 15.7 |

Table 15: **Zero-shot captioning performance on the UCM-captions dataset.**

| Model | BLEU-1 | BLEU-2 | BLEU-3 | BLEU-4 | METEOR | ROGUE_L | CIDEr |
|---|---|---|---|---|---|---|---|
| MiniGPT-v2 (Chen et al., 2023a) | 18.6 | 8.1 | **2.8** | **1.0** | **13.5** | **24.4** | **10.0** |
| GeoChat (Kuckreja et al., 2023) | 19.8 | 6.9 | 0.0 | 0.0 | 8.5 | 19.1 | 4.3 |
| TEOChat | **28.4** | **8.4** | 2.1 | 0.0 | 8.8 | 21.8 | 4.1 |

Table 16: **Zero-shot captioning performance on the Sydney-captions dataset.**

| Model | BLEU-1 | BLEU-2 | BLEU-3 | BLEU-4 | METEOR | ROGUE_L | CIDEr |
|---|---|---|---|---|---|---|---|
| MiniGPT-v2 (Chen et al., 2023a) | 19.9 | 9.2 | 3.9 | 1.6 | **13.7** | **21.4** | **10.7** |
| GeoChat (Kuckreja et al., 2023) | 22.3 | 8.8 | 3.2 | 1.2 | 10.5 | 18.8 | 7.1 |
| TEOChat | **29.2** | **12.9** | **5.5** | 2.5 | 10.4 | 19.5 | 10.1 |

Table 17: **Zero-shot captioning performance on the RSICD dataset.**

| Model | BLEU-1 | BLEU-2 | BLEU-3 | BLEU-4 | METEOR | ROGUE_L | CIDEr |
|---|---|---|---|---|---|---|---|
| MiniGPT-v2 (Chen et al., 2023a) | 18.7 | **8.8** | **3.6** | **1.5** | **14.0** | **23.1** | **12.5** |
| GeoChat (Kuckreja et al., 2023) | 21.6 | 8.7 | 2.5 | 1.0 | 10.0 | 19.8 | 8.4 |
| TEOChat | **22.1** | 8.5 | 3.2 | 1.1 | 9.8 | 20.7 | 10.0 |

Table 18: **Average zero-shot captioning performance across the three datasets.**

## I.3 SINGLE IMAGE GROUNDING RESULTS

We evaluate TEOChat's performance on single image grounding (Table 19). TEOChat outperforms MiniGPTv2 overall (+5.3%) and outperforms GeoChat in detecting large objects (+2.2%), but underperforms GeoChat across the other grounding tasks (-5.5% overall) which we suspect is due to the lower spatial resolution (224 vs. 504) required to handle temporal inputs. Qualitatively, TEOChat tends to make similar correct predictions and errors as GeoChat, but is a little less precise. To investigate the effect of lower spatial resolution on GeoChat's performance, we evaluate GeoChat on 224x224 images (after performing bicubic interpolation of the positional embeddings to account for the change in spatial resolution). This results in huge drops (20-40% absolute drop) in grounding performance across all tasks, potentially suggesting a high sensitivity to input positional information on the grounding tasks.

| Model | Small | Medium | Large | Single-object grounding | Multi-object grounding | [refer] | [grounding] | Overall |
|---|---|---|---|---|---|---|---|---|
| MiniGPT-v2 (Chen et al., 2023a) | 14.3 | 37.3 | 54.7 | 38.0 | 13.1 | 33.1 | 18.3 | 32.0 |
| GeoChat (Kuckreja et al., 2023) | **26.4** | **50.4** | 63.1 | **50.6** | **22.8** | **45.1** | **27.5** | **43.8** |
| GeoChat 224 x 224 | 1.1 | 4.0 | 16.3 | 6.3 | 2.3 | 5.4 | 4.6 | 5.3 |
| TEOChat | 16.3 | 43.0 | **65.5** | 43.1 | 19.1 | 38.2 | 26.2 | 37.3 |

Table 19: **Single EO image grounding performance (Acc@0.5).** We evaluate all tasks using axis-aligned bounding boxes. All metrics are defined as in Kuckreja et al. (2023).

## I.4 TEMPORAL SEQUENCE SAMPLING ABLATION

We run an ablation to measure the impact of the random samples of 8 images taken when the image sequences are longer than 8 images on fMoW RGB and fMoW Sentinel (Table 20).

| Run | fMoW RGB | fMoW Sentinel |
|---|---|---|
| 1 | 75.1 | 45.5 |
| 2 | 75.2 | 45.3 |
| 3 | 75.1 | 45.6 |
| 4 | 75.1 | 45.4 |
| 5 | 75.2 | 45.4 |

Table 20: **Impact of random samples of 8 images from long image sequences on fMoW RGB and fMoW Sentinel validation set performace**. There is minimal variance (0.1-0.2%) from different random samples.

## I.5 ADDITIONAL TEMPORAL QUALITATIVE RESULTS

Examples of TEOChat's responses on canonical temporal tasks are shown in Figure 6.

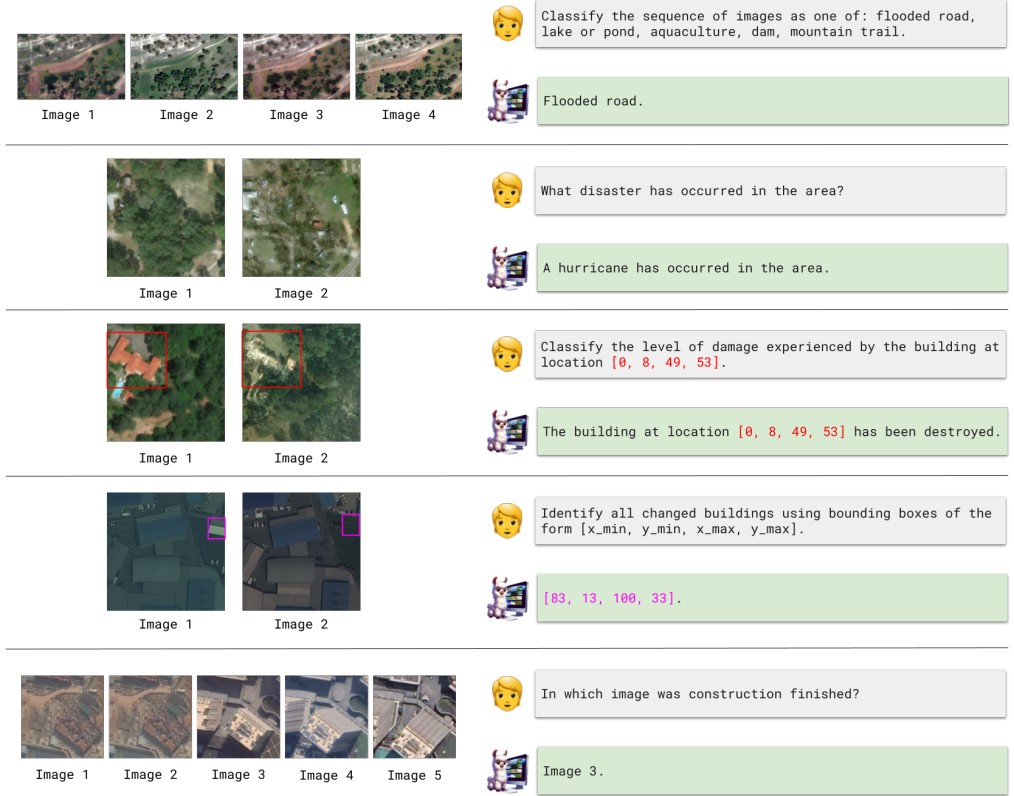

Figure 6: **Qualitative examples of responses produced by TEOChat on temporal tasks.**

### I.6    SINGLE IMAGE QUALITATIVE RESULTS

Examples of responses output by TEOChat on various single EO image tasks are shown in Figure 7, and examples of TEOChat responses on single EO image captioning tasks are provided in Figure 8.

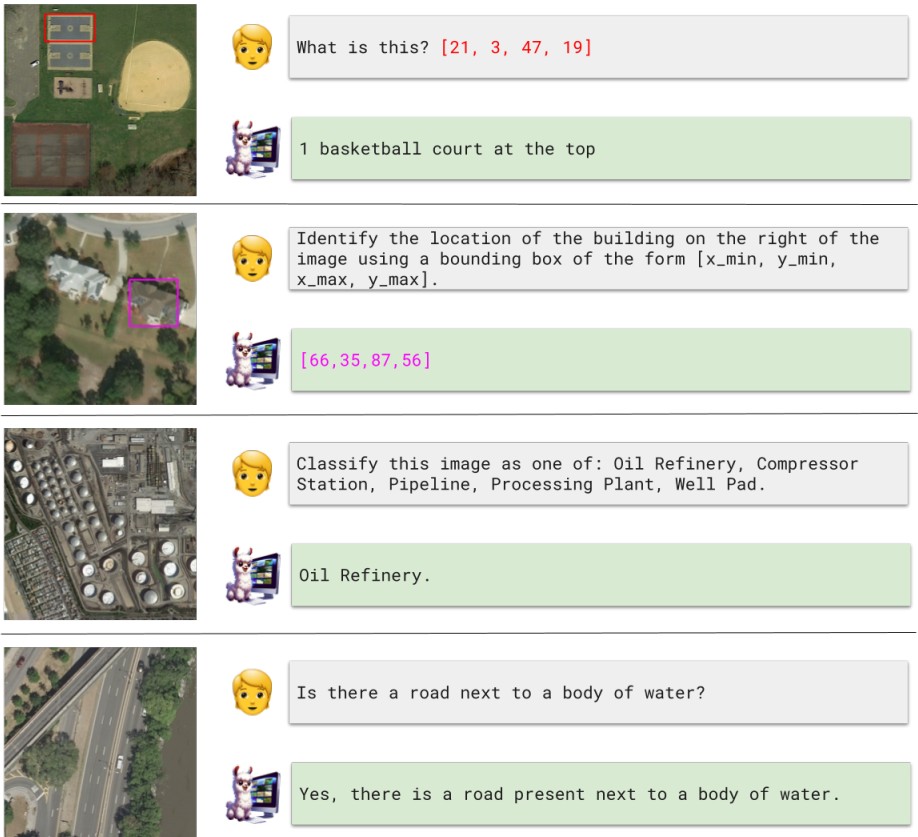

Figure 7: **Examples of TEOChat's responses on single image tasks.**

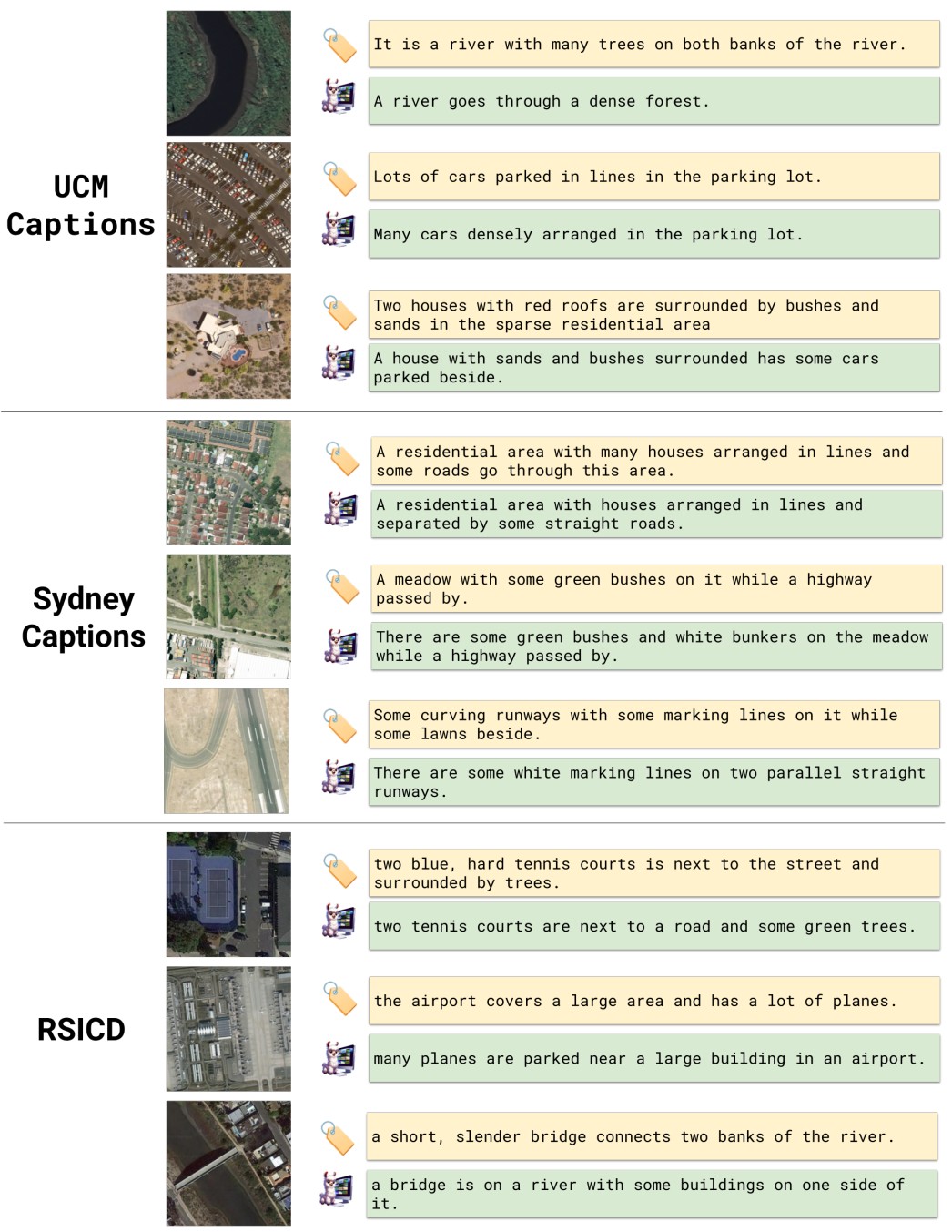

Figure 8: **Examples of TEOChat's responses on the three single image EO captioning datasets after fine-tuning following Hu et al. (2023).**

## J  BROADER IMPACTS

We show that TEOChat is capable of temporal reasoning, achieving strong performance on several real world tasks like building damage detection. Automated, generalist methods for processing temporal EO data like TEOChat have the potential to lead to significant, positive societal impacts. For example, we believe that such methods could be deployed at scale on sequences of Earth observation data to automatically derive information, ranging from disaster response to urban development monitoring. This analysis is typically conducted by experts acquiring data on the ground or by man-

ually inspecting UAV/satellite imagery, which becomes impractical to do in large areas which may require several thousands of high resolution images to capture. Such technology could, for example, aid government agencies and humanitarian organizations to acquire information much more rapidly, which is key for disaster response (Khan et al., 2023). We hope our work can inspire more work on developing vision and language assistants for important temporal EO tasks and lead to helpful new technologies that can benefit society.

We highlight a few potential negative societal impacts of TEOChat. First, TEOChat can automatically extract information from Earth observation imagery, which may lead to privacy concerns, especially as the imagery achieves higher and higher spatial resolution. Second, TEOChat, like other large multimodal models, does not always produce accurate outputs, sometimes hallucinating information or objects in the images, which has the potential to mislead users if misused. It is also possible but unlikely that fine-tuning on TEOChatlas may affect safeguards employed in Llama 2. Ongoing work on mitigating hallucinations in LLMs (Tonmoy et al., 2024) and LMMs (Zhong et al., 2024), as well as work addressing safety of LLMs (Huang et al., 2024), both have the potential to translate to LMMs for Earth observation, but while these approaches continue to mature, users should check outputs for important applications.

## K   AUTHOR CONTRIBUTIONS

1. **Jeremy Andrew Irvin**: Developed the initial research idea, formulated instruction-following tasks, prepared datasets, implemented training experiments and evaluation code, designed and conducted training experiments, drafted the paper.

2. **Emily Ruoyu Liu, Joyce Chuyi Chen, Ines Dormoy**: Prepared datasets, helped design architecture and formulate instruction-following tasks, implemented evaluation code, drafted and edited the paper.

3. **Jinyoung Kim**: Prepared data, edited the paper.

4. **Samar Khanna**: Helped design architecture, designed and conducted training experiments, edited the paper.

5. **Zhuo Zheng**: Supervised the project, helped design architecture and formulate instruction-following tasks, edited the paper.

6. **Stefano Ermon**: Co-developed the research idea, supervised the project, edited the paper.

