# OpenReview forum: "TEOChat: A Large Vision-Language Assistant for Temporal Earth Observation Data"
_ICLR.cc/2025/Conference — ICLR 2025 Poster_

### Official Review · Reviewer_3yvB · 2024-10-20

**Soundness:** 3
**Presentation:** 3
**Contribution:** 3
**Rating:** 8
**Confidence:** 5

**Summary:**

The paper presents TEOchat, a vision-language model (VLM) explicitly designed for the temporal sequences of EO data. While existing temporal sequences models focus on natural images/videos and the EO VLMs focus mainly on single-images, TEOchat fills a gap by being the first to manage temporal EO data. Authors propose a new dataset called TEOchatlas with 554k examples of various single-image and temporal tasks instructions. TEOChat performs well in tasks such as change detection and temporal reasoning, outperforming both generalist and specialist VLMs designed for EO tasks.

**Strengths:**

* A VLM tailored for temporal EO data addresses an important gap in current literature. Prior VLMs could only work with either natural video/multi-images or single EO images.
* The dataset extends GeoChat datasets to the temporal domain. The temporal component of the TEOchatlas dataset is comprised of 4 datasets (fMoW, xBD, S2Looking, QFabric) that are converted to the following temporal tasks: Temporal Scene Classification, Change Detection, Spatial Change Referring Expression, Change QA, Region-based Change QA, Temporal Referring Expression and Region-based Temporal QA.
* The model is comprehensively evaluated across multiple datasets and tasks. TEOChat consistently performs well on both generalist and specialist models on tasks such as change detection, temporal scene classification, and change question answering.  Its performance on zero-shot tasks is also significant.
* Ablations done on the type of projector initialization, its finetuning startegy, and choice of image/video encoder are comprehensive and justify the design choices.

**Weaknesses:**

* Similar to video models like Video-LLaVa, the authors limit the sequences to a maximum of 8 images. For longer sequences, how this restriction can affect the model results?
* TEOchat randomly sample 8 images without replacement from longer sequences, the choice of samples selected can significantly change the results. Can the authors report variance in results due to different random samplings?
* The statement "TEOChat also performs comparably to or outperforms a specialist model trained to perform specific tasks" seems quite a strong claim given the proposed model performs quite low in certain cases: e.g., xBD Loc. (IoU) and QFabric [2 and 5 images].
* Table 1, I believe the GeoChat model is evaluated zero-shot given it cannot be trained with multiple images? This should be clarified clearly in Table 1 that the comparison is between a zero-shot model evaluation (not built for multiple images) and TEOchat that is specifically trained with multiple images. Similarly, as Table 2 shows, the VideoLLava is also evaluated zero-shot in Table 1, and compared with the version that is finetuned on the TEOchatlas dataset, I do not think its a fair comparison and authors should clearly mention that they are comparing zero-shot models with a supervised model.
* Contribution 2 gives the impression that the 554k instruction set is this paper's contribution while over 55% of it is included from previous work GeoChat. It would be reasonable to clarify this and change the claim accordingly.
* The model uses interleaved image identifiers among the visual tokens in the sequence input to the LLM (Figure 3). This differs from prior methods for modeling temporal sequences of natural images which commonly concatenate the tokens and leads to significant gains as per ablation study. However, it is not clear why the image identifier would help in bi-temporal datasets like xBD and S2Looking, since tagging two images should not make much of a difference. Can the authors explain this?

**Questions:**

* A number of open VLMs now allow the processing of multiple images. Can the authors evaluate such options e.g., Qwen2VL, InternVL2 on temporal EO tasks explored in the paper?
* Some recent parallel works (e.g., CDchat https://arxiv.org/html/2409.16261v1) have also explored extending EO VLMs for change detection. Authors are encouraged to explain differences with them (although this is an Arxiv work, and comparing them is not mandatory).
* The VLM responses are textual, the metrics used for eval are accuracy, IoU and F1 measures. Its not clear how the textual outputs are parsed to obtain object coordinates, outputs classes etc. How robust is this process to avoid any errors due to incorrect parsing?
* How much the choice of low-image resolution affect the performance on localization tasks like detection when the objects are quite small. Authors use all the GeoChat data for training, including referring and grounding instructions, but do not compare single-image grounding tasks. It will be good to understand how the choice of vision encoder affects the performance on these tasks.
* Why GPT4 and Gemini models were evaluated/compared on only two tasks? Analysis on the kind of cases where these models fail will be important.

---

> ### Author Response · Authors · 2024-11-19
>
> We really appreciate the reviewer’s thoughtful feedback on our work. Based on that feedback:
> 1. We have run or plan to run additional experiments, including:
> - **Measuring the variance in results from sampling 8 images on fMoW**. We find minimal (0.1-0.2%) variance.
> - **Investigating the impact of image references on bitemporal tasks**. We find it is the delineation of visual tokens between images that explains most of the improvement.
> - **Evaluating open-source multi-image models (Qwen2VL, InternVL2) on the temporal EO tasks**. Results TBD.
> - **Evaluating TEOChat on single image grounding tasks**. Results TBD.
> - **Analyzing failure cases of GPT4o and Gemini-1.5 Pro**. Results TBD.
>
> We will share results from the remaining experiments as they finish.
>
> 2. We have also made changes to the writing in the revised paper, including:
> - Softening the claims about comparisons to specialist models.
> - Clarifying the comparison to GeoChat and Video-LLaVA.
> - Clarifying the amount of TEOChatlas which includes temporal tasks.
> - Explaining differences between our work and the concurrent CDChat work.
>
> Our responses to your individual comments are below.
>
> ---
>
> ### Weaknesses
> > Similar to video models like Video-LLaVa, the authors limit the sequences to a maximum of 8 images. For longer sequences, how this restriction can affect the model results?
>
> The reviewer makes a valid observation about sequence length. The vast majority of EO tasks use sequences with fewer than 8 images due to infrequent high resolution captures of specific locations, and have variable time intervals between captures. Consequently, EO temporal sequences exhibit much less redundancy compared to natural videos, so TEOChat’s ability to perform well on 8 image sequences addresses the practical needs of most EO applications. We note that across all the datasets we explore in this work, fMoW is the only dataset with sequence lengths longer than 8 images, with 87% of the dataset having sequences with 8 or fewer images. Furthermore, our hardware prevents us from running inference with sequence lengths longer than 8 images, but if there is another experiment we could run that would answer your question, please let us know.
>
> ---
>
> > TEOchat randomly sample 8 images without replacement from longer sequences, the choice of samples selected can significantly change the results. Can the authors report variance in results due to different random samplings?
>
> This is an interesting point. We have run an experiment to measure the variance in performance due to the random samples on fMoW, which is the only dataset in our work with sequences containing more than 8 images. Please see the table below.
>
> We find very little variance (0.1 - 0.2%) from different random samplings. We have added this to the paper (Table 12).
>
> |Run|fMoW RGB|fMoW Sentinel|
> |-|-|-|
> |1|75.1|45.5|
> |2|75.2|45.3|
> |3|75.1|45.6|
> |4|75.1|45.4|
> |5|75.2|45.4|
>
> ---
>
> > The statement "TEOChat also performs comparably to or outperforms a specialist model trained to perform specific tasks" seems quite a strong claim given the proposed model performs quite low in certain cases: e.g., xBD Loc. (IoU) and QFabric [2 and 5 images].
>
> This is a fair point, and we have softened the claims throughout the paper accordingly. For example, for that specific statement we have changed it to:
>
> “TEOChat also performs comparably to or outperforms several specialist models trained to perform specific tasks.”
>
> ---
>
> > Table 1, I believe the GeoChat model is evaluated zero-shot given it cannot be trained with multiple images? This should be clarified clearly in Table 1 that the comparison is between a zero-shot model evaluation (not built for multiple images) and TEOchat that is specifically trained with multiple images. Similarly, as Table 2 shows, the VideoLLava is also evaluated zero-shot in Table 1, and compared with the version that is finetuned on the TEOchatlas dataset, I do not think its a fair comparison and authors should clearly mention that they are comparing zero-shot models with a supervised model.
>
> This is another fair point. We have made this more clear in the Table 1 caption in the revised version. Still, we believe it is a meaningful comparison, as the construction of a fine-tuning dataset is one of our contributions and is key to the capabilities demonstrated by TEOChat.
>
> ---
>
> > Contribution 2 gives the impression that the 554k instruction set is this paper's contribution while over 55% of it is included from previous work GeoChat. It would be reasonable to clarify this and change the claim accordingly.
>
> We agree that this would benefit from clarification, and have changed it accordingly in the revised version:
>
> “TEOChatlas contains 245,210 temporal EO examples spanning dozens of temporal instruction-following tasks as well as 308,861 single EO examples from GeoChat_Instruct, totalling 554,071 examples and 1,244,393 images.”

---

> > ### Author Response · Authors · 2024-11-19
> >
> > >The model uses interleaved image identifiers among the visual tokens in the sequence input to the LLM (Figure 3). This differs from prior methods for modeling temporal sequences of natural images which commonly concatenate the tokens and leads to significant gains as per ablation study. However, it is not clear why the image identifier would help in bi-temporal datasets like xBD and S2Looking, since tagging two images should not make much of a difference. Can the authors explain this?
> >
> > This is an interesting observation. Our hypothesis is that clearly delineating which visual tokens come from which image is helpful, even in the bitemporal case. We tested this by running an experiment on the bitemporal datasets where we remove "Image 1" but not "Image 2", and the inverse. Please see the table below.
> >
> > Including `Image 2` leads to much larger performance gains than `Image 1` almost attaining the performance achieved by including both references, suggesting it is in fact the delineation between visual tokens of each image that matters.  We have added this result to the revised paper (Table 13).
> >
> > |Image Reference Strategy|xBD Dmg Cls.|S2Looking Det.|
> > |-|-|-|
> > |No References|44.7|25.5|
> > |Only `Image 1`|47.5|25.8 |
> > |Only `Image 2`|49.2|34.5|
> > |Both `Image 1` and `Image 2`|50.0|34.5|
> >
> > ---
> >
> > ### Questions
> >
> > > A number of open VLMs now allow the processing of multiple images. Can the authors evaluate such options e.g., Qwen2VL, InternVL2 on temporal EO tasks explored in the paper?
> >
> > This is a great suggestion, and we are working on including performance on the temporal EO tasks using a few open source models. We will report the results here once they are done.
> >
> > ---
> >
> > > Some recent parallel works (e.g., CDchat https://arxiv.org/html/2409.16261v1) have also explored extending EO VLMs for change detection. Authors are encouraged to explain differences with them (although this is an Arxiv work, and comparing them is not mandatory).
> >
> > While CDChat is an arXiv work and uploaded past the upload deadline which ICLR considers concurrent work (July 1, 2024), we still agree that explaining differences from our work is valuable, so we have added a comparison to the related work section in our revised paper:
> >
> > “Importantly, no prior work has developed VLMs for temporal sequences for EO except for CDChat, a concurrent work that fine-tunes LLaVA-1.5 on two bitemporal change detection datasets. In contrast, our work builds upon the more recent VideoLLaVA architecture, enabling the handling of sequences with up to eight images. Additionally, we develop a public instruction-following dataset including many different single image tasks as well as temporal tasks from four EO datasets, broadening the scope and capabilities of the model.”
> >
> > ---
> >
> > > The VLM responses are textual, the metrics used for eval are accuracy, IoU and F1 measures. Its not clear how the textual outputs are parsed to obtain object coordinates, outputs classes etc. How robust is this process to avoid any errors due to incorrect parsing?
> >
> > This is a great question. We design the prompts to make the desired output format very clear. While this does not guarantee correct parsing in every case, we observed fewer than 0.1% errors across all evaluations we conducted. We have added this information to the revised paper (Appendix Section E).
> >
> > ---
> >
> > > How much the choice of low-image resolution affect the performance on localization tasks like detection when the objects are quite small. Authors use all the GeoChat data for training, including referring and grounding instructions, but do not compare single-image grounding tasks. It will be good to understand how the choice of vision encoder affects the performance on these tasks.
> >
> > This is a fair point. We are evaluating TEOChat on single-image grounding tasks (broken down by object size) to investigate how TEOChat’s detection performance on lower resolution images compares to that of GeoChat on higher resolution images. We will share the results after.
> >
> > ---
> >
> > > Why GPT4 and Gemini models were evaluated/compared on only two tasks? Analysis on the kind of cases where these models fail will be important.
> >
> > GPT4 and Gemini-1.5 Pro are expensive to run inference with several images (both in the input, and with examples used for in-context learning), so we were only able evaluate them on the bitemporal datasets with a reasonable number of images. We include a statement on this in Section 5.4:
> >
> > “We could not evaluate the models on fMoW or QFabric (which are larger both in size and sequence length) due to cost.”
> >
> > We agree analysis on the failure cases of these models would be interesting. We are working on that and will report back with our results after.

---

> > > ### Author Response · Authors · 2024-11-24
> > >
> > > In [the next revision](https://openreview.net/pdf?id=pZz0nOroGv), we have added the following based on your feedback:
> > >
> > > **1. A comparison to two open-source multi-image VLMs (Qwen2VL-7B and InternVL2-4B).**
> > >
> > > We find InternVL2-4B and Qwen2VL-7B-Instruct underperform both proprietary models, as well as TEOChat by a large margin on xBD damage classification (-38.2 F1, -32.7 F1 respectively) and S2Looking change detection (-25.8 F1, -23.7 F1 respectively). We have updated Table 4 accordingly, pasting below for convenience:
> > >
> > > |Model|xBD Dmg Cls.|S2Looking Det.|
> > > |-|-|-|
> > > |InternVL2-4B|11.8|7.8|
> > > |Qwen2VL-7B-Instruct|17.3|9.9|
> > > |Gemini 1.5 Pro|35.8|16.5|
> > > |GPT-4o |38.3|21.5|
> > > |**TEOChat**|**50.0**|**33.6**|
> > >
> > > ---
> > >
> > > **2. An evaluation of TEOChat on single image grounding tasks.**
> > >
> > > TEOChat outperforms MiniGPTv2 overall (+5.3%) and outperforms GeoChat in detecting large objects (+2.2%), but underperforms GeoChat across the other grounding tasks (-5.5% overall) which we suspect is due to the lower spatial resolution required to handle temporal inputs. Qualitatively, TEOChat tends to make similar correct predictions and errors as GeoChat, but is a little less precise. Again we note TEOChat has additional temporal capabilities that GeoChat does not.
> > >
> > > We have added these results to the paper (Table 12), pasting below for convenience:
> > >
> > > |Model|Small|Medium|Large|Single-object grounding|Multi-object grounding|[refer]|[grounding]|Overall|
> > > |-|-|-|-|-|-|-|-|-|
> > > |MiniGPTv2|14.3|37.3|54.7|38.0|13.1|33.1|18.3|32.0|
> > > |GeoChat|**26.4**|**50.4**|63.1|**50.6**|**22.8**|**45.1**|**27.5**|**43.8**|
> > > |**TEOChat**|16.3|43.0|**65.5**|43.1|19.1|38.2|26.2|37.3|
> > >
> > > ---
> > >
> > > **3. An analysis of the failure cases of GPT4o and Gemini-1.5 Pro.**
> > >
> > > We find:
> > > - On S2Looking change detection, both Gemini1.5 Pro and GPT4o tend to predict that no buildings have changed when there is change, with GPT4o making this error more often. When they do predict changed buildings, they commonly do not capture all of the change and the bounding boxes do not localize the buildings accurately.
> > > - On xBD damage classification, both GPT4o and Gemini1.5 Pro almost always predict buildings as `No Damage` or `Destroyed`, with Gemini1.5 Pro correctly classifying more destroyed buildings than GPT4o.
> > >
> > > We also inspected InternVL2-4B and Qwen2VL-7B-Instruct predictions and observed that they failed to produce any meaningful bounding boxes in S2Looking change detection, and predicted No Damage in almost every case for xBD damage classification, with Qwen correctly identifying destroyed buildings in the more obvious examples.
> > >
> > > We have added this analysis to the paper (Section 5.4).
> > >
> > > If you could please let us know if we have addressed your concerns, we would really appreciate it.

---

> ### Comment · Reviewer_3yvB · 2024-11-27
>
> I want to thank the authors for their responses to my comments. I am positive about the paper overall and would vote for acceptance. The data and analyses presented in this work are interesting, as this is the first effort in temporal sequence processing with LMMs. I would suggest the authors to include new analysis and results in the revised version when they are available.

---

> > ### Author Response · Authors · 2024-11-27
> >
> > Thank you very much for your positive feedback on our work. The new results and analysis we have conducted during this discussion period are included in [the newest revision](https://openreview.net/pdf?id=pZz0nOroGv).
> >
> > Your comments have helped to improve our paper substantially. We would really appreciate if you would consider updating your score to reflect this improvement. Thank you again for your review of our work.

---

> > > ### Author Response · Authors · 2024-11-28
> > >
> > > We sincerely appreciate you increasing your rating of our work. The newest version is [here](https://openreview.net/pdf?id=pZz0nOroGv) for your reference. Thank you very much again for all of your help improving our paper.

---

### Official Review · Reviewer_qA8u · 2024-11-03

**Soundness:** 2
**Presentation:** 2
**Contribution:** 2
**Rating:** 6
**Confidence:** 4

**Summary:**

In this paper, the authors develop a novel vision and language assistant (TEOChat) to engage in conversations about temporal sequences of earth observation data. Their extensive experiments show that TEOChat can perform a variety of spatial and temporal reasoning tasks, achieving comparable or better performance than specialist models.

**Strengths:**

+ The paper develops the first vision-language model (VLM) for temporal EO data and introduces the first temporal EO dataset for multimodal instruction-tuning, expanding the application scope for VLMs.

+ The proposed method shows impressive zero-shot performance on several datasets, and outperforms foundation models (i.e., GPT-4o and Gemini-1.5 Pro) for modeling sequences of images.

**Weaknesses:**

- While contributing a novel application and a new dataset is encouraging, the technical contribution of this paper is insufficient.  This  paper primarily focuses on employing Low-Rank Adaptation (LoRA) to fine-tune a LLaVA-1.5-like architecture, which allows the model to handle both temporal and Earth Observation (EO) data. This paper could benefit from a more in-depth analysis of why such a design for the framework was chosen.

-  Although the authors mention that their approach achieves impressive performance on various datasets (refer to Table 1), the analysis of these results lacks specificity. It is recommended to present relevant visualizations to demonstrate the hypothesis.

- While the study provides a valuable analysis of the proposed approach against existing benchmarks, it would be enhanced by incorporating more recent specialist methods into the comparative analysis.

- There are some typos in the paper. For example, Line 106: “To align the the EO imagery”, double “the”.

**Questions:**

My concerns are listed in the ''Weaknesses".

---

> ### Author Response · Authors · 2024-11-19
>
> We sincerely appreciate the reviewer’s feedback on our work. Based on that feedback:
> 1. **We have added a comparison to SOTA specialists on the datasets**.
> 2. **We will be including several more qualitative examples demonstrating TEOChat’s capabilities more specifically**.
> 3. **We have fixed a few typos and improved some language throughout the paper.**
>
> Our responses to your individual comments are below.
>
> ---
>
> > While contributing a novel application and a new dataset is encouraging, the technical contribution of this paper is insufficient. This paper primarily focuses on employing Low-Rank Adaptation (LoRA) to fine-tune a LLaVA-1.5-like architecture, which allows the model to handle both temporal and Earth Observation (EO) data. This paper could benefit from a more in-depth analysis of why such a design for the framework was chosen.
>
> We appreciate the reviewer’s feedback. We would like to emphasize that our work is the first to develop a vision and language assistant for temporal EO data, enabling the model to handle many more tasks and datasets than previous foundation models for remote sensing data all within a single framework. The model is high performing across these tasks, which is nontrivial to accomplish as demonstrated by our experimental ablations. We believe this has important implications on future research in foundation models for remote sensing, and these findings could also be valuable to researchers developing VLMs for image sequences in other domains.
>
> We agree that motivation for the design of the framework is necessary. We include multiple paragraphs motivating the design of TEOChat in Section 4.1 and we report several ablations justifying design decisions empirically in Section 5.2. Specifically, we motivate the design choices for:
> - _VLM architecture_: an adapted Video-LLaVA [1] architecture, extending LLaVA-1.5 [2].
> - _Weight initialization_: Video-LLaVA outperforms LLaVA-1.5 in our ablations.
> - _Fine-tuning strategy_: LoRA for memory/retaining pretrained knowledge, frozen vision encoder for memory as is common among EO VLMs [3-6], projector tuning which outperforms freezing in our ablations.
> - _Prompt design_: including image references, which outperforms their exclusion at train-time and test-time in our ablations.
>
> If the reviewer still has concerns about how we motivate our design of the approach, please let us know.
>
> [1] Video-LLaVA: Learning United Visual Representation by Alignment Before Projection. EMNLP 2024.
>
> [2] Improved Baselines with Visual Instruction Tuning. CVPR 2024 Highlight.
>
> [3] RSGPT: A Remote Sensing Vision Language Model and Benchmark. arXiv 2023.
>
> [4] GeoChat: Grounded Large Vision-Language Model for Remote Sensing. CVPR 2024.
>
> [5] LHRS-Bot: Empowering Remote Sensing with VGI-Enhanced Large Multimodal Language Model. ECCV 2024.
>
> [6] VHM: Versatile and Honest Vision Language Model for Remote Sensing Image Analysis. arXiv 2024.
>
> ---
>
> > Although the authors mention that their approach achieves impressive performance on various datasets (refer to Table 1), the analysis of these results lacks specificity. It is recommended to present relevant visualizations to demonstrate the hypothesis.
>
> We will be adding several more qualitative examples of TEOChat’s capabilities across different tasks to show specific examples of its strong performance. If the reviewer has any other suggestions for more specificity or visualizations, please let us know.
>
> ---
>
> > While the study provides a valuable analysis of the proposed approach against existing benchmarks, it would be enhanced by incorporating more recent specialist methods into the comparative analysis.
>
> Thank you for the suggestion. We include the most recent specialist results for QFabric and a comparable, recent specialist for fMoW RGB in Table 1. In the revised version, we now include a comparison to the state-of-the-art (SOTA) performance on fMoW RGB in the results text [8], which TEOChat attains close performance to (75.1% vs. 79.2%). We also now include a comparison to SOTA on S2Looking detection [9] and xBD localization [10] in the results text, and find that TEOChat considerably underperforms them in pixel-based F1 score. This is largely because these two bitemporal change detection models output segmentation masks, whereas TEOChat is constrained to output boxes. As we state in the Conclusion, further improving object localization is an important direction for future work.
>
> [8] ExPLoRA: Parameter-Efficient Extended Pre-Training to Adapt Vision Transformers under Domain Shifts. arXiv 2024.
>
> [9] A New Learning Paradigm for Foundation Model-based Remote Sensing Change Detection. IEEE TGRS 2024.
>
> [10] ChangeMamba: Remote Sensing Change Detection with Spatio-Temporal State Space Model. IEEE TGRS 2024.
>
> ---
>
> > There are some typos in the paper. For example, Line 106: “To align the the EO imagery”, double “the”.
>
> Thank you for pointing out this issue. We have fixed this typo and a few others.

---

> > ### Comment · Reviewer_qA8u · 2024-11-26
> >
> > Authors well addressed some of my concerns. I still have one suggestion:
> >
> > Authors should provide more discussions of motivations in Section 1, which will make this manuscript to be more self-contained and convinced.

---

> > > ### Author Response · Authors · 2024-11-27
> > >
> > > Thank you very much for your additional feedback. We are very glad to hear we have addressed concerns of yours.
> > >
> > > ---
> > >
> > > > Authors should provide more discussions of motivations in Section 1, which will make this manuscript to be more self-contained and convinced.
> > >
> > > We appreciate this suggestion. We have provided more motivation for framework design in the contributions in Section 1 which you can see in [the newly revised paper](https://openreview.net/pdf?id=pZz0nOroGv).
> > >
> > > If the reviewer believes we should add any further motivation in the paper, please let us know and we would be happy to make further changes.
> > >
> > > ---
> > >
> > > **We thank you again for your comments which have helped to substantially improve our paper.**
> > >
> > > **If you have any remaining concerns, please let us know and we would be happy to discuss further. If you do not have additional concerns, we would be very grateful if you would consider updating your score to reflect this. Thank you again for your review of our work.**

---

> > > > ### Comment · Reviewer_qA8u · 2024-11-27
> > > >
> > > > Thanks for authors' responses. I raise the score for recommendation.
> > > >
> > > > However, authors can further highlight the motivation in Sec. 1 in the revised manuscript. Considering the limited pages, authors can take some experiments into the appendix.

---

> > > > > ### Author Response · Authors · 2024-11-28
> > > > >
> > > > > > Thanks for authors' responses. I raise the score for recommendation.
> > > > >
> > > > > We really appreciate you increasing your score, and thank you again for your assistance in making our paper stronger.
> > > > >
> > > > > ---
> > > > >
> > > > > > However, authors can further highlight the motivation in Sec. 1 in the revised manuscript. Considering the limited pages, authors can take some experiments into the appendix.
> > > > >
> > > > > We have expanded the motivation in Sec. 1 in [the revised manuscript](https://openreview.net/pdf?id=pZz0nOroGv), trimming the experiments and dataset section to make space. If the reviewer has any additional comments, please let us know.

---

> ### Author Response · Authors · 2024-11-24
>
> Based on your feedback, in the next revision we have **added more qualitative examples to show TEOChat’s generalization capabilities which are not present in the baselines**. We now highlight them in Figure 4 in the main text, and moved the canonical task examples previously in Figure 4 to the Appendix (Figure 7 in Appendix I.5). The new examples show that TEOChat exhibits generalization to tasks not explicitly represented in its training dataset. Notably, it can compose its spatial and temporal reasoning capabilities, for example identifying objects only regionally annotated in the single image tasks (like airplanes and cars) within different images in the sequence.
>
> If you could please let us know if we have addressed your concerns, we would be very grateful.

---

### Official Review · Reviewer_e7d5 · 2024-11-04

**Soundness:** 3
**Presentation:** 2
**Contribution:** 2
**Rating:** 5
**Confidence:** 3

**Summary:**

The paper introduces a instruction tuning dataset for temporal earth observation and a corresponding fine-tuned model trained on this data.

**Strengths:**

S1. Applications around geo-spatial data are important and have potential for great societal impact.

S2. Applying VLMs to video data is a timely direction.

S3. Additional evidence that state-of-the-art fine-tuning methods are broadly applicable to specific target domains.

**Weaknesses:**

W1. The introduction often reads as a related work section. Consider moving some of the contextualization of the field in the related work section and focus a bit more on the core story and contributions in the intro.

W2. While there is a long history of geo-spatial data in computer vision, the paper contribution does strike me a little narrow in scope.

W3. Methodological novelty. The paper applies a fairly straightforward fine-tuning recipe. While the paper does not oversell the method, this combined with the narrow scope (W2) do make me a bit concerned.

W4. What are some broader applications of these agents? Is it possible to deploy these agents at scale to understand patterns in data that would be intractable for humans to otherwise deduce themselves?

W5. The task presented seem like important capabilities but not particularly difficult for specialized models to solve. How do baseline classifiers, referring expression, change detection, etc. algorithms perform on the tasks. What generalization capabilities can be expected in the proposed model that are not present in these baselines?

**Questions:**

Please address the weaknesses above.

---

> ### Author Response · Authors · 2024-11-19
>
> We are very grateful for the reviewer’s insightful comments on our work. Based on the comments:
> 1. **We have split off a separate related works section from the introduction**.
> 2. **We have added more discussion on applicability and scalability of TEOChat**.
> 3. **We will be adding more qualitative examples demonstrating TEOChat’s generalization**.
>
> Our responses to your individual comments are below.
>
> ---
>
> > W1. The introduction often reads as a related work section. Consider moving some of the contextualization of the field in the related work section and focus a bit more on the core story and contributions in the intro.
>
> Thank you for your feedback on the introduction. As you can see in [the revised paper](https://openreview.net/pdf?id=pZz0nOroGv), we have moved many references into a separate related work section (placed in Appendix Section A due to space limitations) and now focus on the core story and contributions in the intro.
>
> ---
>
> > W2. While there is a long history of geo-spatial data in computer vision, the paper contribution does strike me a little narrow in scope.
>
> > W3. Methodological novelty. The paper applies a fairly straightforward fine-tuning recipe. While the paper does not oversell the method, this combined with the narrow scope (W2) do make me a bit concerned.
>
> We appreciate the reviewer’s feedback. We would like to emphasize that our work is the first to develop a vision and language assistant for temporal EO data, enabling the model to handle many more tasks and datasets than previous foundation models for remote sensing data all within a single framework. The model is high performing across these tasks, which is nontrivial to accomplish as demonstrated by our experimental ablations. We believe this has important implications on future research in foundation models for remote sensing, and these findings could also be valuable to researchers developing VLMs for image sequences in other domains.
>
> ---
>
> > W4. What are some broader applications of these agents? Is it possible to deploy these agents at scale to understand patterns in data that would be intractable for humans to otherwise deduce themselves?
>
> This is an interesting question which we believe deserves discussion in the paper. We include two paragraphs on broader impacts in the Appendix, and have added more about these points there:
>
> “Automated, generalist methods for processing temporal EO data like TEOChat have the potential to lead to significant, positive societal impacts. For example, we believe that such methods could be deployed at scale on sequences of Earth observation data to automatically derive information, ranging from disaster response to urban development monitoring. This analysis is typically conducted by experts acquiring data on the ground or by manually inspecting UAV/satellite imagery, which becomes impractical to do in large areas which may require several thousands of high resolution images to capture. Such technology could, for example, aid government agencies and humanitarian organizations to acquire information much more rapidly, which is key for disaster response [1].”
>
> [1] A systematic review of disaster management systems: approaches, challenges, and future directions. Land 2023.
>
> ---
>
> > W5. The task presented seem like important capabilities but not particularly difficult for specialized models to solve. How do baseline classifiers, referring expression, change detection, etc. algorithms perform on the tasks. What generalization capabilities can be expected in the proposed model that are not present in these baselines?
>
> This is a great question. We have compared to specialist baseline models which can only perform a single task (classification, change detection, and region-based question answering), and have found that TEOChat performed comparably to or outperforms many of them.
>
> Furthermore, TEOChat can perform tasks specified through natural language, making its uses much more broad than any specialist. We have seen TEOChat exhibit interesting generalization to single and temporal image tasks not explicitly represented in the dataset because of this natural language interface. We will add qualitative examples of this to the next version of the paper and let you know when we do.

---

> ### Author Response · Authors · 2024-11-24
>
> In [the next revision](https://openreview.net/pdf?id=pZz0nOroGv), based on your feedback we have **added more qualitative examples to show TEOChat’s generalization capabilities which are not present in the baselines**. We now highlight them in Figure 4 in the main text, and moved the canonical task examples previously in Figure 4 to the Appendix (Figure 7 in Appendix I.5). The new examples demonstrate that TEOChat exhibits generalization capabilities not explicitly represented in its training dataset. Notably, TEOChat can compose its spatial and temporal reasoning abilities, for example identifying objects only regionally annotated in the single image tasks (like airplanes and cars) within different images in the sequence.
>
> If you could please let us know if we have addressed your concerns, we would be very thankful.

---

> > ### Author Response · Authors · 2024-11-28
> >
> > We thank the reviewer again for their comments which have helped to greatly improve our paper. The most recent version is [here](https://openreview.net/pdf?id=pZz0nOroGv). If you have no further concerns, we would greatly appreciate if you would consider adjusting your score accordingly. Thank you again.

---

> > > ### Comment · Reviewer_e7d5 · 2024-11-29
> > >
> > > Thanks for all of your effort here! Upon revision, I am opting to keep my original score. I am still concerned about:
> > > * The narrow scope of the contributions
> > > * Evaluation of model generalization. While some qualitative examples are a reasonable response for purposes of a rebuttal, ultimately it would be nice to see quantitative generalization on held-out tasks. I think adding these measurements would be incredibly nice for a re-submission!
> > >
> > > Thanks again for all of the work!

---

> > > > ### Author Response · Authors · 2024-11-29
> > > >
> > > > Thank you very much for your additional feedback on our work!
> > > >
> > > > ---
> > > >
> > > > > The narrow scope of the contributions
> > > >
> > > > We are thankful for the feedback on the perceived scope of our contributions. It is important to note that ICLR routinely publishes papers similar to our work that introduce new datasets aimed at fostering novel modeling capabilities (see some examples from ICLR 2024 [1-9]), many of which are focused on specific domains [3-9].  Like these papers, our contribution is twofold: first, we introduce a unique paired image-text dataset for temporal sequences of EO images which will enable the development of temporal instruction-following methods, and holds potential for other image-text tasks like contrastive approaches and text-image generative models. Second, we develop a new model that demonstrates impressive temporal instruction-following performance as validated on more than 10 datasets, achieving capabilities not previously exhibited by existing models. Because of this, we think our contributions are a strong fit for ICLR.
> > > >
> > > > ---
> > > >
> > > > > Evaluation of model generalization. While some qualitative examples are a reasonable response for purposes of a rebuttal, ultimately it would be nice to see quantitative generalization on held-out tasks. I think adding these measurements would be incredibly nice for a re-submission!
> > > >
> > > > Thank you for recognizing the value of the new qualitative examples we have added.  We note that we already include quantitative generalization results to held-out tasks (ABCD and CDVQA), on which TEOChat demonstrates impressive performance, considerably outperforming the baseline and approaching the performance of specialist models trained on and customized to each task (Table 3, Section 5.3). We also present quantitative performance of TEOChat’s generalization to three held-out single image datasets in the main text, two of which include tasks not explicitly represented in the training data (Table 5, Section 5.5).
> > > >
> > > > We are not aware of any other temporal EO datasets that would allow us to quantitatively evaluate TEOChat’s generalization to new spatiotemporal tasks like the ones presented in Figure 4. If the reviewer has a suggestion for another dataset that would provide additional value or novel insights into TEOChat’s capabilities, we would be happy to explore this.
> > > >
> > > > ---
> > > >
> > > > **Thank you again for your help reviewing our work. We are happy to further discuss the work through the end of the discussion period.**
> > > >
> > > > ---
> > > >
> > > > [1] InternVid: A Large-scale Video-Text Dataset for Multimodal Understanding and Generation. _ICLR 2024_.
> > > >
> > > > [2] LMSYS-Chat-1M: A Large-Scale Real-World LLM Conversation Dataset. _ICLR 2024_.
> > > >
> > > > [3] Remote Sensing Vision-Language Foundation Models without Annotations via Ground Remote Alignment. _ICLR 2024_.
> > > >
> > > > [4] OpenWebMath: An Open Dataset of High-Quality Mathematical Web Text. _ICLR 2024_.
> > > >
> > > > [5] CircuitNet 2.0: An Advanced Dataset for Promoting Machine Learning Innovations in Realistic Chip Design Environment. _ICLR 2024_.
> > > >
> > > > [6] Large-scale Training of Foundation Models for Wearable Biosignals. _ICLR 2024_.
> > > >
> > > > [7] Towards Foundational Models for Molecular Learning on Large-Scale Multi-Task Datasets. _ICLR 2024_.
> > > >
> > > > [8] Mol-Instructions: A Large-Scale Biomolecular Instruction Dataset for Large Language Models. _ICLR 2024_.
> > > >
> > > > [9] BrainLM: A foundation model for brain activity recordings. _ICLR 2024_.

---

### Official Review · Reviewer_7dgE · 2024-11-05

**Soundness:** 2
**Presentation:** 3
**Contribution:** 1
**Rating:** 1
**Confidence:** 5

**Summary:**

The work attempts to develop the first vision-language model for temporal earth observation data (EOD). When an EOD is prompted with temporal images and NLP, the proposed model can perform tasks that need temporal reasoning,

**Strengths:**

This paper introduces TEOChat, which is the first unified model for temporal earth observation (EO) data
The authors create a novel EO dataset

**Weaknesses:**

It would be more convincing if the authors could provide more evidence on the diversity and representativeness of the dataset, as well as fine-tuning results on supervised datasets.

Why only a comparison with GeoChat is made, whereas many other similar works are available?

What is the performance for other RS captioning datasets such as UCM-CAPTIONS and SYDNEY-CAPTIONS datasets which are mentioned in RSGPT[1].

It will also be important to showcase the performance w.r.t other performance measures such as recall etc

[1] RSGPT: A Remote Sensing Vision Language Model and Benchmark

In addition, the architecture and concept are very similar to Video-ChatGPT which has never been discussed or compared. Also, there are RS-based VLM which target change detection (https://github.com/lzw-lzw/awesome-remote-sensing-vision-language-models) which will be important to compare.

**Questions:**

As mentioned in the weakness

---

> ### Author Response · Authors · 2024-11-19
>
> We sincerely thank the reviewer for providing helpful feedback on our work. Based on that feedback, we have added the following to [our revised paper](https://openreview.net/pdf?id=pZz0nOroGv):
> 1. **A comparison to works besides GeoChat.** TEOChat demonstrates similar or better single image performance compared to other single image only approaches on six of the seven tasks.
> 2. **RS captioning results on UCM-captions, Sydney-Captions, and RSICD.** TEOChat outperforms specialists and performs competitively with generalist single image EO models on the larger two of the captioning datasets (UCM-captions and RSICD).
>
> Point-by-point responses follow.
>
> ---
> > It would be more convincing if the authors could provide more evidence on the diversity and representativeness of the dataset, as well as fine-tuning results on supervised datasets.
>
> We appreciate the reviewer's point. In the Appendix, Figure 5 plots the geographic distribution of TEOChatlas, showing it is highly diverse and representative of many areas around the world, and Table 7 shows it spans several sensors/spatial resolutions and tasks.  Is there any other information we could include to further prove its diversity/representativeness?
>
> Regarding fine-tuning, we now include results from fine-tuning on 3 captioning datasets (as done in RSGPT), described below. While we agree fine-tuning on other datasets would be interesting, we feel this is out of scope of our study, which aims to develop a model that used for many different tasks without the need for fine-tuning. However, we will include code in our open-source repo to make it easier for users to fine-tune TEOChat on other datasets.
>
> ---
> > Why only a comparison with GeoChat is done, whereas many other similar works are available?
>
> We compared TEOChat to GeoChat as it is the most directly comparable model: it uses a similar architecture and is trained on a subset of the TEOChatlas dataset. However, we agree with the reviewer that comparing to other works would still be beneficial, so we have added results from other models to Table 5. See the table below.
>
> TEOChat shows strong performance across the single image tasks, achieving the highest performance on LRBEN presence (+0.5%) and composition (+0.7%), close to highest on LRBEN Rural/Urban (-1.0%), and highest zero-shot on HRBEN Presence (+3%).  It notably underperforms LHRS-Bot and VHM on AID (-10.4-10.8%), which we believe is likely due to the inclusion of several more LULC classification datasets in their training sets. We reemphasize that both of those models were only designed for single image tasks, whereas TEOChat can also perform temporal tasks. We added these results to the paper (Table 5), pasting for convenience:
>
> |Model|AID|UCMerced|LRBEN|||HRBEN||Average|
> |-|-|-|-|-|-|-|-|-|
> ||||**Pres.**|**Comp.**|**Rural/Urban**|**Pres.**|**Comp.**||
> |Video-LLaVA|52.4|46.5|55.8|65.1|61.0|64.5|66.8|58.5|
> |RSVQA|-|-|_87.5_|_81.5_|_90.0_|_90.4_|_88.2_|-|
> |RSGPT|-|-|_91.2_|_91.7_|_94.0_|_90.9_|_90.0_|-|
> |LHRS-Bot|91.3|-|_88.5_|_90.0_|_89.1_|_92.6_|_92.5_|-|
> |VHM|**91.7**|-|_91.1_|_92.0_|**_95.0_**|64.0|**83.5**|-|
> |GeoChat|72.0|84.4|_91.1_|_90.3_|_94.0_|58.5|83.2|79.9|
> |**TEOChat**|80.9|**86.3**|**_91.7_**|**_92.7_**|_94.0_|**67.5**|81.1|**83.4**|
>
> We _italicize_ results from fine-tuning on the corresponding training set. All other results are zero-shot. We **bold** the highest zero-shot score, except LRBEN which all models but Video-LLaVA are fine-tuned on.
>
> ---
> > What is the performance for other RS captioning datasets such as UCM-CAPTIONS and SYDNEY-CAPTIONS datasets which are mentioned in RSGPT[1].
>
> This is an interesting question. Prior works evaluating on these captioning datasets include them in the training dataset, so we have run experiments fine-tuning TEOChat on UCM-Captions, Sydney-Captions, and RSICD following RSGPT’s procedure. Please see Tables 9-11 in the revised Appendix, which are not pasted here for brevity.
>
> Compared to both single image EO specialist and generalist models, we find that TEOChat achieves the highest performance across multiple metrics on UCM-captions and comparable performance to RSGPT on RSICD. On Sydney-captions, TEOChat underperforms RSGPT, but qualitatively we find TEOChat’s captions are high quality. We emphasize that in addition to these single image capabilities, TEOChat has temporal reasoning capabilities that none of these other models have.
>
> ---
> > It will be also important to showcase the performance w.r.t other performance measures such as recall etc
>
> We appreciate the reviewer’s sentiment about performance measures. We used the standard, commonly reported metrics for each dataset, and tried to include a diverse set of metrics across classification and detection tasks including accuracy, F1, and IoU. We can report F1 performance broken down by precision and recall. Does the reviewer have suggestions for other metrics that would help holistically measure performance?

---

> > ### Comment · Reviewer_7dgE · 2024-11-26
> > **Further comments**
> >
> > Thank you for the revision. After reading the other review and the rebuttals, there are still certain queries.
> >
> > The results of GeoChat in Table 12 are much better; however, the explanation added is not very convincing.
> >
> > Table 9-11 results on GeoChat are missing.
> >
> > Also, the motivation needs to be better aligned w.r.t the end tasks.

---

> > > ### Author Response · Authors · 2024-11-27
> > >
> > > Thank you very much for your additional feedback. We have incorporated your feedback into [the next version of the revised paper](https://openreview.net/pdf?id=pZz0nOroGv), specifically:
> > > 1. **Obtaining GeoChat results on the captioning tasks.**
> > > 2. **Adding more motivation for the methodology and for the tasks in Figure 4.**
> > > 3. **Changing language regarding the grounding results in Table 12.**
> > >
> > > Our individual responses to each point are below.
> > >
> > > ---
> > >
> > > > The results of GeoChat in Table 12 are much better; however, the explanation added is not very convincing.
> > >
> > > We appreciate the reviewer’s point about an explanation for Table 12. This is not feasible to ablate as we are unable to train TEOChat with images larger than 224x224 resolution on our hardware due to GPU memory constraints. To investigate the effect of lower spatial resolution on GeoChat's performance, we tried running inference with GeoChat on 224x224 images (after correctly interpolating positional embeddings to account for the change in spatial resolution), but doing so results in huge drops (20-40% absolute drop) in grounding performance across all tasks.
> > >
> > > We note that except for grounding, TEOChat has matched or surpassed GeoChat performance across all single image benchmark tasks, and has additional temporal capabilities. We believe it is not too surprising that TEOChat’s grounding capabilities are less strong than GeoChat’s given the 2.25x higher spatial resolution used by GeoChat. This is supported by the fact that TEOChat outperforms GeoChat in the detection of large objects (+2.4%) but underperforms considerably on small objects (-8.1%). Still, this is a limitation we acknowledge a few times in the paper (in Results Section 5.1 and the Conclusion), and we also now specify in the Abstract of the revised paper that TEOChat is superior than GeoChat on specific types of single image tasks, rather than talking about its superiority broadly.
> > >
> > > If the reviewer still has concerns about these results, or has a suggestion for an experiment we could run or further changes we could make to the paper, please let us know.
> > >
> > > ---
> > >
> > > > Table 9-11 results on GeoChat are missing.
> > >
> > > This is a great point. GeoChat does not report results on these captioning datasets, so we have fine-tuned their model using their training code and training procedure on each of the datasets. We have added the results to Tables 9-11 in the newly revised paper. TEOChat's performance exceeds GeoChat's across all metrics on all three datasets except for METEOR on UCM-captions, on which it slightly underperforms (-0.4%).
> > >
> > > ---
> > >
> > > > Also, the motivation needs to be better aligned w.r.t the end tasks.
> > >
> > > We thank the reviewer for pointing this out. We are interpreting “end tasks” to mean the generalization to new tasks presented in Figure 4, which we agree could be better motivated and contextualized in the paper. To address this, we have now added more motivation for the tasks in Figure 4 by adding it as a finding at the top of Section 5 and a separate subsection discussing the findings (Section 5.6). We have also added further motivation for the design of our approach to achieve the aims of the paper.
> > >
> > > If this is not what the reviewer meant or if the reviewer still believes the motivation should be improved, please let us know.
> > >
> > > ---
> > >
> > > **Thank you again for the feedback. Your comments have helped to improve our paper substantially. If you have any remaining concerns, please let us know and we would be happy to discuss further.**

---

> ### Comment · Reviewer_7dgE · 2024-11-27
> **Further insight**
>
> Thanks for the revision.
>
> 1. the results for the proposed method on the caption datasets (RSICD and Sydney) are not the best. some explanation and showcasing average results on all datasets might add value.  It will also be important to showcase the zero short results.
>
> **If any of these analyses can exhibit the superiority of the proposed method, I will be in favour of the paper and raise my rating.**
>
> 2. Ok about this
> 3. Might add the results of GeoChat on 224x224 as another row to the table.

---

> > ### Author Response · Authors · 2024-11-28
> >
> > Thank you for the further comments. We have made further changes to [the newest version](https://openreview.net/pdf?id=pZz0nOroGv) based on the feedback, described below.
> >
> > ---
> >
> > > the results for the proposed method on the caption datasets (RSICD and Sydney) are not the best. some explanation and showcasing average results on all datasets might add value. It will also be important to showcase the zero short results.
> >
> > These are good recommendations. We have added:
> > 1. **Average results on the three captioning datasets (Table 14).** On average, TEOChat outperforms all approaches (including GeoChat and two specialists) except for RSGPT, which we explain below.
> > 2. **Zero-shot results on the three captioning datasets (Tables 15-17).** We now evaluate against GeoChat and MiniGPT-v2 in the zero-shot setting, but we could not evaluate against RSGPT as their model is not open source. TEOChat performs similarly to or outperforms GeoChat on average across all metrics, with notable improvements on the largest, more difficult captioning dataset (RSICD). Interestingly, MiniGPT-v2 achieves impressive zero-shot performance on these datasets, which we explain below.
> > 3. **Added explanation of captioning results (Appendix Section I.2).** First, we believe RSGPT’s strong fine-tuning performance as well as MiniGPT-v2’s strong zero-shot and fine-tuning performance are both largely due to the inclusion of a higher proportion of captioning datasets in their training data, whereas GeoChat and TEOChat have more scene classification and question answering tasks. Second, these three captioning datasets have several known issues which make their use as benchmarks suboptimal, including limited vocabulary and very short descriptions, and sometimes even misspelled words and grammatical errors. We have included example labels in Figure 8 and add this explanation to the results in the new paper.
> >
> > ---
> >
> > > Might add the results of GeoChat on 224x224 as another row to the table.
> >
> > This is a great suggestion. We have added another row to Table 13 and discuss the results in Appendix Section I.3 in the revised version.
> >
> > ---
> >
> > **Thank you again for your feedback, and please let us know if you have any more suggestions.**

---

> > > ### Comment · Reviewer_7dgE · 2024-11-28
> > > **further insight**
> > >
> > > 1. The average results on the caption are not better than RSGPT
> > > 2. In zero-short also the results are not promising
> > >
> > > Need some more detailed insight
> > >
> > > 1. Also in Table 5 RSGPT is much better for VQA ~92 is average whereas Teochat is ~83, results on scene classification have not been provided for RSGPT.
> > >
> > > 2. It will be important to showcase how RSGPT works on multi-image VLMs in Tables 4 and 5
> > >
> > > Otherwise, the contribution of the paper is not very promising.

---

> > > > ### Comment · Reviewer_7dgE · 2024-11-28
> > > > **Results are different**
> > > >
> > > > Also, the results reported for the caption dataset in RSGPT and what is reported in TeoChat for RSGPT are very different. Please clarify?
> > > >
> > > > Also for GeoChat for UCM wrt BLEU-4, ROUGE-L, METEOR results are  80.7, 79.1, 75.4  and as reported in TeoChat is 68.5,  78.1, 44.4. Please clarify?

---

> > > > > ### Author Response · Authors · 2024-11-28
> > > > >
> > > > > > Also, the results reported for the caption dataset in RSGPT and what is reported in TeoChat for RSGPT are very different. Please clarify?
> > > > >
> > > > > We have confirmed the results we have listed in Tables 11-13 for RSGPT match the results listed in Tables III - V in [the RSGPT paper](https://arxiv.org/pdf/2307.15266). If we have misunderstood what the reviewer means, please let us know.
> > > > >
> > > > > ---
> > > > >
> > > > > > Also for GeoChat for UCM wrt BLEU-4, ROUGE-L, METEOR results are 80.7, 79.1, 75.4 and as reported in TeoChat is 68.5, 78.1, 44.4. Please clarify?
> > > > >
> > > > > We are not able to find the numbers you cite for those metrics in our paper, nor are we able to find any results for UCM captions in the GeoChat paper. The results reported in Table 10 in the GeoChat paper are not on one of the three captioning datasets, and are region level captioning results not image-level, so they are not comparable. If the reviewer could share where those numbers are reported, then we can further clarify.

---

> > > > > > ### Comment · Reviewer_7dgE · 2024-11-28
> > > > > > **details on GeoChat**
> > > > > >
> > > > > > UCM is in GeoChat supply: https://github.com/mbzuai-oryx/GeoChat/blob/main/docs/geochat_supp.pdf
> > > > > >
> > > > > > See Table 11 in TEOChat

---

> > > > > > > ### Author Response · Authors · 2024-11-28
> > > > > > >
> > > > > > > Thank you for bringing this to our attention.
> > > > > > >
> > > > > > > Since these results were uploaded 30 minutes ago, there were no existing results for us to report in our paper, therefore we fine-tuned GeoChat ourselves using the training code from the GeoChat repo.
> > > > > > >
> > > > > > > However, the differences in results that we report and the ones newly reported in the GeoChat supplement are very likely due to differences in training procedure. Given the code for fine-tuning GeoChat on the captions datasets has not been released, and there is not much detail about the fine-tuning procedure in the supplement, it is difficult for us to deduce the exact reason for the differences.
> > > > > > >
> > > > > > > Furthermore, the METEOR results reported may be inflated as the scores are much higher than all previous scores on the two datasets, and artificially higher METEOR scores can be caused by using an incorrect implementation (using NLTK instead of Java): https://github.com/nltk/nltk/issues/2655
> > > > > > >
> > > > > > > We would be happy to discuss any of these points further. Thank you again.

---

> > > > > > > > ### Comment · Reviewer_7dgE · 2024-11-28
> > > > > > > > **In GeoChat supply is in cvpr24 website**
> > > > > > > >
> > > > > > > > Thanks, it is also available at: https://openaccess.thecvf.com/content/CVPR2024/supplemental/Kuckreja_GeoChat_Grounded_Large_CVPR_2024_supplemental.pdf
> > > > > > > >
> > > > > > > > Whatever the fact my point is results that are reported do not match. If it is reevaluated and desired results are not achieved then it's a big flag for me. They should be done in the best setting for fair comparison.

---

> > > > > > > > > ### Author Response · Authors · 2024-11-28
> > > > > > > > >
> > > > > > > > > Thank you for sharing this other link. We were not aware of this supplemental material as it did not appear in our search results and is still not present in the arXiv version of the paper. We appreciate the reviewer bringing it to our attention.
> > > > > > > > >
> > > > > > > > > We agree that these specific results are worth investigating further. The code to fine-tune GeoChat on the caption datasets is not available, so it is not currently possible to reproduce the results and determine the reason for the difference. However, we will try to work with the GeoChat authors to investigate this discrepancy.
> > > > > > > > >
> > > > > > > > > We also reemphasize that these few supplemental single image captioning results are not a main contribution of our work, which is focused on new temporal EO capabilities.
> > > > > > > > >
> > > > > > > > > Thank you again for continuing this discussion, and please let us know if you have any other concerns.

---

> > > > > > > > > > ### Comment · Reviewer_7dgE · 2024-11-28
> > > > > > > > > > **Conclusion**
> > > > > > > > > >
> > > > > > > > > > Through the progress in the discussion, I arrived at the point that the paper is not ready for publication. Because many experiments are not been completed which I pointed out. Although there is good massage but needs more experimental validation.  Moreover, there is a discrepancy when the results are compared to SOTA.  Hence the contribution of the paper is not justified and it is not possible to complete then by the time we arrive at the end of the discussion phase. Hence, I decided to lower my score for the recommendation of the paper. If the experiments are done we can see a better submission in the future conference.

---

> > > > > > > > > > > ### Author Response · Authors · 2024-11-28
> > > > > > > > > > >
> > > > > > > > > > > To try to facilitate further discussion, we would like to share our understanding of the reviewer’s points and why we strongly disagree with the reviewer’s conclusion.
> > > > > > > > > > >
> > > > > > > > > > > The reviewer claims that the reason they believe our paper is not ready for publication is that many experiments they have pointed out have not been completed. However, we have completed all of their requested experiments during the discussion period. We are enumerating the experiments suggested by the reviewer below:
> > > > > > > > > > >
> > > > > > > > > > > **1. “Fine-tuning results on supervised datasets” and “performance for other RS captioning datasets such as UCM-CAPTIONS and SYDNEY-CAPTIONS”. [Reviewer Comment](https://openreview.net/forum?id=pZz0nOroGv&noteId=dP63HiznTj)**
> > > > > > > > > > >
> > > > > > > > > > > We ran experiments to obtain TEOChat’s fine-tuning results on both UCM-captions and Sydney-captions in addition to a third dataset, RSICD (Tables 11-13 in the newest version, [Author Comment](https://openreview.net/forum?id=pZz0nOroGv&noteId=z6oxRcIF2f)). We reemphasize these results are not central to the paper’s main claim as the datasets are not temporal EO benchmarks.
> > > > > > > > > > >
> > > > > > > > > > > **2. “Table 9-11 results on GeoChat are missing.” [Reviewer Comment](https://openreview.net/forum?id=pZz0nOroGv&noteId=SuRN0nqthU)**
> > > > > > > > > > >
> > > > > > > > > > > We ran experiments to obtain GeoChat’s fine-tuning results on all three datasets (Tables 11-13, [Author Comment](https://openreview.net/forum?id=pZz0nOroGv&noteId=QwxC3sXzUM)). The reviewer has just brought it to our attention that these results do not match the results reported in the GeoChat supplement, but we have no way of understanding these differences as their caption fine-tuning, inference, and evaluation code is not open source, and their fine-tuning procedure is not described with sufficient detail. There is also evidence to believe at least some of the metrics reported in the GeoChat supplement are in fact incorrect, given the METEOR scores look far too high in comparison with prior methods ([Author Comment](https://openreview.net/forum?id=pZz0nOroGv&noteId=LnlpMjYADu)).
> > > > > > > > > > >
> > > > > > > > > > > **3.  “the results for the proposed method on the caption datasets (RSICD and Sydney) are not the best. some explanation and showcasing average results on all datasets might add value. It will also be important to showcase the zero short results.” [Reviewer Comment](https://openreview.net/forum?id=pZz0nOroGv&noteId=fboxKCyrpM)**
> > > > > > > > > > >
> > > > > > > > > > > We added average results on all datasets (Table 14) and ran experiments to obtain zero-shot results for MiniGPTv2, GeoChat, and TEOChat (Tables 15-18, [Author Comment](https://openreview.net/forum?id=pZz0nOroGv&noteId=x9ekjXruao)).
> > > > > > > > > > >
> > > > > > > > > > > ---
> > > > > > > > > > >
> > > > > > > > > > > We also want to highlight that:
> > > > > > > > > > > - These experiments requested by the reviewer are supplementary, single image results that deviate from the main contribution of our paper, which is to develop a VLM for temporal EO data. Our goal is not to improve upon single EO image performance, and these captioning experiments are additional single image results we added on top of the several other single image results we present in the main text.
> > > > > > > > > > > - After running these additional experiments and making several changes to the paper requested by the reviewer, they decided to change their rating to the lowest possible score.
> > > > > > > > > > >
> > > > > > > > > > > **We kindly request that the other reviewers carefully read the full discussion with reviewer 7dgE and form their own opinions.**

---

> > > > ### Author Response · Authors · 2024-11-28
> > > >
> > > > Thank you very much for your further comments.
> > > >
> > > > ---
> > > > >1. The average results on the caption are not better than RSGPT
> > > > >2. In zero-short also the results are not promising.
> > > >
> > > > We hypothesize that these are both partially due to training dataset composition differences, and reemphasize that these caption benchmarks have several known issues including limited vocabulary and very short descriptions, and sometimes even misspelled words and grammatical errors. In our revised paper we have cautioned readers to avoid inferring too much from results on these datasets.
> > > >
> > > > We also note that our major contribution is not improved single image performance but TEOChat's new temporal capabilities. While we are unable to evaluate RSGPT's temporal capabilities as it is not open source, it likely does not have strong temporal capabilities as no temporal tasks are in its training data.
> > > >
> > > > ---
> > > >
> > > > >1. Also in Table 5 RSGPT is much better for VQA ~92 is average whereas Teochat is ~83, results on scene classification have not been provided for RSGPT.
> > > >
> > > > This is because unlike TEOChat, RSGPT was trained on the corresponding training set for the HR VQA tasks which we have displayed using _italics_ in Table 5. Both models were trained on the LR VQA tasks making it a more fair comparison, and TEOChat outperforms RSGPT on those tasks.
> > > >
> > > > ---
> > > >
> > > > >2. It will be important to showcase how RSGPT works on multi-image VLMs in Tables 4 and 5
> > > >
> > > > RSGPT is not open source, so we are unable to evaluate its performance on multi-image tasks.
> > > >
> > > > ---
> > > > We hope this clarifies the concerns raised by the reviewer. If you have any additional comments, please let us know and we are happy to discuss further.

---

> ### Comment · Reviewer_7dgE · 2024-11-29
> **Official coment by reviwer 7dgE**
>
> I can not appreciate the last reply from the authors:
>
> 1. if some experiments are requested that exhibit poor results does that showcase the novelty of the proposed method?
> 2. I Would request to go through GitHub and other sources of a paper before criticizing,
>
>
> 3: ** -For this comment "There is also evidence to believe at least some of the metrics reported in the GeoChat supplement are in fact incorrect"**- How do you conclude this? I will request  **-ACs, SACs, and PCs**. for your attention on this. It is very illegitimate comment made by the author to prove something. **_Is this the purpose of a discussion?_**

---

> ### Author Response · Authors · 2024-11-29
>
> We acknowledge the concerns raised by the reviewer and respectfully offer the following clarifications:
> 1. As we have stated, these newly added single image captioning results are **not core to the contributions or claims of our paper, and there are known issues with these benchmarks that necessitate caution when inferring from results**. Still, TEOChat achieves results reasonably competitive with other single image methods on average in both the fine-tuning and zero-shot settings according to our results (Table 14, Table 18, Figure 8).
> 2. We have gone extensively through the GeoChat GitHub repository, and did not find the supplemental material, only available separately in the CVPR repository, until the reviewer shared the newly uploaded pdf on GitHub. However, as we stated before, these supplemental results are **only relevant to the new captioning experiments this reviewer requested during the discussion period**. We will resolve any discrepancies in these results to the best of our ability.
> 3. We have not concluded that the presented results are certainly incorrect, just that the drastic improvement in METEOR score puts the results into question. To be more specific, on RSICD, recent methods achieve METEOR scores ranging from 23.0 to 30.1 (see RSGPT), GeoChat reports a METEOR score of 49.3, and our computed numbers are 28.9 for TEOChat and 28.0 for GeoChat. On UCM captions, recent methods achieve scores ranging from 36.5 to 42.2, GeoChat reports a METEOR score of 75.4, and our computed numbers are 43.9 for TEOChat and 44.4 for GeoChat. The numbers we compute are much more similar to the ranges reported in RSGPT. Should the GeoChat fine-tuning, inference, and evaluation code for the captions datasets be released in the future, we would be happy to conduct a more thorough investigation.
>
> As our discussion has shifted away from the core aspects of our work, we remain open to discussion about aspects more related to our central claims and contributions. We still greatly appreciate the reviewer’s time and effort to review and discuss our paper, so we thank you again.

---

> > ### Author Response · Authors · 2024-12-04
> >
> > We noticed that the reviewer edited their original post to add two new weaknesses, which we wanted to respond to below:
> >
> > **Comparing to Video-ChatGPT:** Video-ChatGPT underperforms Video-LLaVA across all tested tasks in the Video-LLaVA paper,  so we feel strongly that this is not a necessary comparison.
> >
> > **Comparing to other RS-based VLMs for Change Detection:**  There are no other RS-based VLMs for change detection besides the concurrent work CDChat, which has several key differences as we discuss in the paper, described [here](https://openreview.net/forum?id=pZz0nOroGv&noteId=6YKDE0yna4).
> >
> > Thank you for your additional comments.

---

### Author Response · Authors · 2024-11-19
**Initial Response to Reviewers**

We kindly thank the reviewers for their helpful feedback on our work. We appreciate that they find our approach effective (`7dgE`, `e7d5`, `qA8u`, `3yvB`), our experiments extensive (`qA8u`, `3yvB`), and our dataset novel, filling an important gap in the literature (`7dgE`, `qA8u`, `3yvB`).

The major concerns raised by the reviewers include:
1. Comparing to other single image and multi-image instruction-following approaches.
2. Comparing to state-of-the-art (SOTA) specialist approaches.
3. Adding EO captioning and single image grounding results.
4. Measuring the impact of long sequence subsampling in fMoW and benefit from image references on the bitemporal datasets.
5. Adding more qualitative examples demonstrating TEOChat’s capabilities.

To address reviewer feedback, we have included or plan to include the following in our revision:
1. **A comparison to several single EO image instruction-following models and open-source multi-image instruction-following models.** We find TEOChat also achieves better performance than all single EO image instruction-following models on four of seven tasks, similar performance to the best performing models on two, and underperforms two models on one task, which we attribute to dataset composition differences. We are currently evaluating two open-source multi-image instruction-following models for comparison as well.
2. **A comparison to SOTA specialist approaches.** TEOChat’s performance is impressively close to specialist SOTA on fMoW RGB (75.1% vs. 79.2%), and lower than SOTA on xBD localization and S2Looking detection, which we mostly attribute to TEOChat’s constrained output format of boxes versus segmentation masks.
3. **Additional evaluations on three EO captioning datasets and on single image grounding.** We find TEOChat exceeds the performance of specialists and matches generalist single image EO models on the larger two of the three captioning datasets, and shows strong captioning abilities qualitatively on all three datasets. We are working on getting single image grounding results as well.
4. **New ablations testing the effect of long sequence subsampling in fMoW and investigating the reason why image references benefit bitemporal performance.** We find minimal variance due to the sampling (0.1-0.2%) and that the `Image 2` reference explains most of the performance gains in the bitemporal case (85% on xBD damage classification, 100% on S2Looking detection), suggesting it is primarily the delineation of visual tokens from the images that is beneficial in the bitemporal setting.
5. **Additional qualitative examples of TEOChat’s predictions on various EO tasks.** We will be adding these in the next revision.

As mentioned, we are running a few additional experiments and will follow up with results as we obtain them. In the meantime, we would be very grateful if reviewers could discuss our initial responses to their comments and [the first version of our revised paper](https://openreview.net/pdf?id=pZz0nOroGv), which has changes marked in `red`. Except for a minor error we corrected in our per-pixel F1 implementation that affected a few results, all changes to the paper are described in our reviewer-specific responses below.

Thank you all again for your review of our work.

---

> ### Author Response · Authors · 2024-11-24
>
> We have made [another revision of the paper](https://openreview.net/pdf?id=pZz0nOroGv) to address the remaining reviewer comments. Specifically, as we stated we would do in our previous post, we have added:
> 1. **A comparison to two open-source multi-image VLMs, namely Qwen2-VL and InternVL2 (Table 4).** TEOChat outperforms both Qwen2-VL and InternVL2 by a large margin on xBD damage classification (+38.2 F1, +32.7 F1 respectively) and S2Looking change detection (+25.8 F1, +23.7 F1 respectively).
> 2. **An evaluation of TEOChat on single image grounding tasks (Table 12).** TEOChat outperforms MiniGPTv2 across all grounding tasks (+5.3% overall) and outperforms GeoChat in detecting large objects (+2.2%) but underperforms GeoChat across the other grounding tasks, which we largely attribute to the lower spatial resolution required to handle the extra temporal inputs. We emphasize, however, that TEOChat has additional temporal capabilities that GeoChat does not.
> 3. **An analysis of the failure cases of GPT4o and Gemini-1.5 Pro (Section 5.4).** Gemini-1.5 Pro and GPT4o tend to underpredict change in S2Looking change detection and underestimate building damage in xBD damage classification.
> 4. **More qualitative examples of TEOChat outputs, including instances of generalization (Figure 4 in Results) and example predictions on the captioning datasets (Figure 8 in Appendix Section I.6).** Interestingly, the examples show that TEOChat can compose its spatial and temporal capabilities in ways not explicitly represented in the training dataset.
>
> We would be very grateful if the reviewers could examine these changes (all marked in `red` in [the revised paper](https://openreview.net/pdf?id=pZz0nOroGv)) and let us know if your concerns have been addressed.
>
> Thank you again.

---

> > ### Author Response · Authors · 2024-11-27
> >
> > Based on the recent additional comments, we have uploaded [another revision](https://openreview.net/pdf?id=pZz0nOroGv) with a few more changes:
> > 1. **Adding GeoChat’s captioning results.** TEOChat outperforms GeoChat across all metrics on all three captioning datasets except one metric (METEOR) on one dataset (UCM-captions), for which it slightly underperforms GeoChat (-0.4%). This further demonstrates TEOChat’s strong single image performance, with similar or superior performance to GeoChat across zero-shot land use and land cover classification, visual question answering, and now EO captioning.
> > 2. **More discussion of general motivation for the work.** We add more motivation for TEOChat’s design in Section 1 and better motivate the results in Figure 4.
> > 3. **Language changes based on the grounding results in Table 12.** We soften the claim about the broad superiority of TEOChat over GeoChat on single image tasks in the Abstract. We emphasize that TEOChat has matched or surpassed GeoChat performance across all single image benchmark tasks except grounding, and has additional temporal capabilities. We also believe it is reasonable that TEOChat would underperform GeoChat on grounding given the 2.25x lower spatial resolution of the input images required to handle the extra temporal inputs.
> >
> > We also made a few minor changes to the language throughout the paper to stay within the page limit. All changes are marked in `red`.
> >
> > If the reviewers have additional comments, we would be happy to discuss them further. Thank you again for your very helpful review of our work.

---

> ### Author Response · Authors · 2024-11-28
>
> We have made a few more changes to [the revised paper](https://openreview.net/pdf?id=pZz0nOroGv) based on the recent comments, including:
> 1. **Average results on the three captioning datasets (Table 14).** TEOChat outperforms all approaches (including GeoChat and two specialists) besides RSGPT. We add explanation for this in the paper, described in detail to reviewer `7dgE` below.
> 2. **Zero-shot results on the three captioning datasets and average performance (Tables 15-17).** TEOChat performs similarly to or outperforms GeoChat on average across all metrics, with notable improvements on the largest, more difficult captioning dataset (RSICD).
> 3. **Adding more motivation to Section 1.** We have trimmed extraneous language in the dataset construction and experiments section in order to make room for this additional motivation in Section 1.
>
> We believe our paper has substantially improved because of the feedback, so we are very grateful to the reviewers for their valuable help to make the work stronger. If there are any remaining concerns, we would be happy to continue discussing.

---

### Author Response · Authors · 2024-12-04
**Summary of Paper Changes during Discussion Period**

As the discussion period comes to a close, we want to sincerely thank the reviewers again for their valuable feedback on our work.

Below, we summarize the major changes made to our paper based on the reviewer feedback. All of the changes to the paper are marked in `red` in [our revised paper](https://openreview.net/pdf?id=pZz0nOroGv).

1. **A comparison to additional single EO image instruction-following models (Table 1), SOTA specialists (Section 5.1), and open-source multi-image instruction-following models (Table 4).** TEOChat achieves similar or higher single image performance than all models on all but one task (Table 1), achieves impressively close to SOTA results on fMoW but lower on xBD localization and S2Looking detection (Section 5.1), and considerably higher performance than two open-source multi-image instruction-following models (Table 4).

2. **Additional evaluations on three single EO image captioning datasets (fine-tuning in Tables 11-14, zero-shot in Tables 15-18) and on single image grounding (Table 19).**  TEOChat achieves captioning performance competitive with other single image methods on average quantitatively in both the fine-tuning and zero-shot settings (Table 14, Table 18) and demonstrates strong performance qualitatively (Figure 8). On single image grounding, TEOChat outperforms MiniGPTv2 across all grounding tasks and outperforms GeoChat in detecting large objects but underperforms GeoChat across the other grounding tasks (Table 19), which we largely attribute to the 2.25x lower spatial resolution required to handle the extra temporal inputs. We emphasize that TEOChat has additional temporal capabilities that GeoChat does not.

3. **Additional qualitative examples of TEOChat’s predictions on various EO tasks (Figures 4, 6-8).** Interestingly, the examples demonstrate TEOChat’s ability to compose its spatial and temporal reasoning capabilities in novel ways not explicitly represented in the training set (Figure 4).

4. **New ablations testing the effect of long sequence subsampling in fMoW (Table 20) and investigating the reason why image references benefit bitemporal performance (Table 9).** We find minimal variance due to the sampling (Table 20) and that it is primarily the delineation of visual tokens between the images that makes the image references beneficial in the bitemporal setting (Table 9).

5. **More discussion of general motivation for the work and a separate related works section (Section 1, Appendix Section A).** We add further motivation and context to the Introduction (Section 1), and include a separate related works section (Appendix Section A) to better contextualize our contributions.

We believe these changes have substantially improved our paper, which we are very grateful for. Thank you all again for your thoughtful review of our work.

---

### Meta-Review · Area_Chair_H559 · 2024-12-22

**Metareview:**

The manuscript received ratings of 5, 8, 6, and 1. Reviewers appreciated that the manuscript introduces a model tailored for temporal Earth observation (EO) data along with tailored instruction set for tempora domain in remote sensing, thereby addressesing an important gap in existing literature. Reviewers also rasied several issues in the manuscript including, processing longer sequences (beyond  8 images), certain claims regarding the performance of TeoChat (e.g., xBD Loc. (IoU) and QFabric [2 and 5 images]), evaluating existing models (e.g., Qwen2VL, InternVL2) on temporal EO tasks, and comparison with other remote sensing captioning datasetss such as UCM-Captions and Sydney-Captions. Authors submitted a rebuttal to address the concerns of reviewers. One reviewer remained concerns on single image performance of proposed TeoChat on some datasets. Authors have detailed discussions with the reviewer explaining the results further (e.g., lack of open-source code availability of RSGPT) while also highlighting efficacy of the proposed approach in terms of its temporal capabilities. Two reviewers mentioned that their respective concerns are addressed during the rebuttal and remained positive, while the fourth reviewer remained concer about the narrow scope of the work.  The meta-reviewer believes that the proposed work has merits overall as it goes beyond single image remote sensing understanding to handling temporal Earth observations. Results on temporal Earth observation tasks are promising, while the authors have also provided additional results and comparisons with single EO image instruction-following models in the rebuttal. Given the reviewers comments, rebuttal and the detailed discussions, the recommendation is accept. Authors are strongly encouraged to take into consideration reviewers feedback when preparing the revised manuscript.

**Additional Comments On Reviewer Discussion:**

Reviewers appreciated that the manuscript introduces a model tailored for temporal Earth observation (EO) data along with tailored instruction set for tempora domain in remote sensing, thereby addressesing an important gap in existing literature. Reviewers also rasied several issues in the manuscript including, processing longer sequences (beyond  8 images), certain claims regarding the performance of TeoChat (e.g., xBD Loc. (IoU) and QFabric [2 and 5 images]), evaluating existing models (e.g., Qwen2VL, InternVL2) on temporal EO tasks, and comparison with other remote sensing captioning datasetss such as UCM-Captions and Sydney-Captions. Authors submitted a rebuttal to address the concerns of reviewers. One reviewer remained concerns on single image performance of proposed TeoChat on some datasets. Authors have detailed discussions with the reviewer explaining the results further (e.g., lack of open-source code availability of RSGPT) while also highlighting efficacy of the proposed approach in terms of its temporal capabilities. Two reviewers mentioned that their respective concerns are addressed during the rebuttal and remained positive, while the fourth reviewer remained concer about the narrow scope of the work.  The meta-reviewer believes that the proposed work has merits overall as it goes beyond single image remote sensing understanding to handling temporal Earth observations. Results on temporal Earth observation tasks are promising, while the authors have also provided additional results and comparisons with single EO image instruction-following models in the rebuttal.

---

### Decision · Program_Chairs · 2025-01-22

Accept (Poster)